# Genomic inference of the metabolism and evolution of the archaeal phylum Aigarchaeota

Zheng-Shuang Hua[1], Yan-Ni Qu[1], Qiyun Zhu [2], En-Min Zhou[1], Yan-Ling Qi[1], Yi-Rui Yin[1], Yang-Zhi Rao[1], Ye Tian[1], Yu-Xian Li[1], Lan Liu[1], Cindy J. Castelle[3], Brian P. Hedlund[4,5], Wen-Sheng Shu[6], Rob Knight [2,7,8] & Wen-Jun Li [1,9]

Microbes of the phylum Aigarchaeota are widely distributed in geothermal environments, but their physiological and ecological roles are poorly understood. Here we analyze six Aigarchaeota metagenomic bins from two circumneutral hot springs in Tengchong, China, to reveal that they are either strict or facultative anaerobes, and most are chemolithotrophs that can perform sulfide oxidation. Applying comparative genomics to the Thaumarchaeota and Aigarchaeota, we find that they both originated from thermal habitats, sharing 1154 genes with their common ancestor. Horizontal gene transfer played a crucial role in shaping genetic diversity of Aigarchaeota and led to functional partitioning and ecological divergence among sympatric microbes, as several key functional innovations were endowed by Bacteria, including dissimilatory sulfite reduction and possibly carbon monoxide oxidation. Our study expands our knowledge of the possible ecological roles of the Aigarchaeota and clarifies their evolutionary relationship to their sister lineage Thaumarchaeota.

[1] State Key Laboratory of Biocontrol, Guangdong Key Laboratory of Plant Resources, School of Life Sciences, Sun Yat-Sen University, 510275 Guangzhou, China. [2] Department of Pediatrics, University of California San Diego, La Jolla, CA 92093, USA. [3] Department of Earth and Planetary Science, University of California, Berkeley, Berkeley, CA 94720, USA. [4] School of Life Sciences,  University of Nevada Las Vegas, Las Vegas, NV 89154, USA. [5] Nevada Institute of Personalized Medicine, University of Nevada Las Vegas, Las Vegas, NV 89154, USA. [6] School of Life Sciences, South China Normal University, 510631 Guangzhou, China. [7] Department of Computer Science and Engineering, University of California San Diego, La Jolla, CA 92093, USA. [8] Center for Microbiome Innovation, University of California San Diego, La Jolla, CA 92093, USA. [9] College of Fisheries, Henan Normal University, 453007 Xinxiang, China. These authors contributed equally: Zheng-Shuang Hua, Yan-Ni Qu.  Correspondence and requests for materials should be addressed to W.-J.L. (email: liwenjun3@mail.sysu.edu.cn)

Recent advancements in metagenomics and single-cell techniques have enabled researchers to obtain a glimpse of the genomic information and genetic diversity of major lineages of Bacteria and Archaea that have eluded microbial cultivation[1–4]. One such group is the archaeal lineage Aigarchaeota. Members of the Aigarchaeota are widely distributed in terrestrial and subsurface geothermal systems and marine hydrothermal environments[3,5–9]. To date, eight non-redundant Aigarchaeota single-amplified genomes (SAGs) and metagenome-assembled genomes (MAGs) exist in the IMG and NCBI RefSeq databases. However, CheckM[10] estimates that only one is >90% complete. Phylogenetic analysis of conserved, single-copy core genes showed that Aigarchaeota is closely related to Crenarchaeota, Korarchaeota and Thaumarchaeota[3,7], forming the "TACK" superphylum[11]. As a sister lineage, Thaumarchaeota has been largely studied with respect to their importance in nitrogen cycling, particularly chemolithotrophic ammonia oxidation[12–14]. In contrast, little is known about Aigarchaeota's ecological role or evolutionary history. It is still controversial whether Aigarchaeota represents a new phylum or a subclade of phylum Thaumarchaeota[7,9,15,16].

Here, by integrating genome-resolved metagenomics and comparative genomics, we aim to address the gaps in understanding the origin and evolutionary history of the Aigarchaeota, and their roles in biogeochemical cycling. Our results reveal the facultative or strictly anaerobic lifestyle of Aigarchaeota with the ability to oxidize sulfide to gain energy for growth. Evolutionary genomic analyses of Thaumarchaeota and Aigarchaeota suggest both two phyla evolved from hot habitats and they share a large proportion of gene families with their last common ancestor. Adaptation to different habitats led to the functional differentiation of Aigarchaeota and Thaumarchaeota, with the latter invading a wide diversity of non-thermal environments. Genes acquired horizontally from Euryarchaeota and Firmicutes played a significant role in the functional diversification of Aigarchaeota

in hot spring habitats. This study represents a significant advancement in our understanding of the genomic diversity and evolution of Aigarchaeota.

## Results and discussion

**Aigarchaeota genomes recovered from metagenomes.** Metagenome sequencing was performed on two hot spring sediment samples collected from Tengchong county of Yunnan, China (Fig. 1a). Metagenome assembly of sequencing data and genome binning based on tetra-nucleotide frequencies and coverage patterns were conducted, resulting in six near-complete Aigarchaeota genomic bins (Table 1). One bin was obtained from Gumingquan (GMQ, pH 9.3 and temperature 89 °C)[17], and five from another pool named Jinze (JZ, pH 6.5 and temperature 75 °C)[18] (Supplementary Table 1). The genome sizes of the obtained bins range from 1.09 Mbp to 1.65 Mbp (averagely 1.4 Mbp). They encode an average of 1384 genes with an average gene length of 905 bp. Genomes are well curated and genome quality was evaluated using CheckM[10], indicating the high-quality of the genomic bins with genome completeness ranging from 97 to 99% and nearly no contamination of other genome fragments with 16S rRNA and tRNA (>18) are detectable[19] (Table 1; Supplementary Fig. 1). A concatenated alignment of 16 ribosomal proteins was used for maximum likelihood tree construction (Fig. 1b). A phylogenetic tree with high bootstrap support reveals that the six genomes are located in distinct lineages. A 16S rRNA-based phylogenetic tree indicates that they represent three groups with 94% as a threshold (Supplementary Fig. 2). GMQ bin_10 and JZ bin_10 are the close neighbors sharing 97% of the amino acid identity (AAI) (Supplementary Figs. 3, 4). JZ bins_28 shows 92% AAI to the published genome of "*Candidatus* Caldiarchaeum subterraneum" (Supplementary Fig. 3). The only detected Thaumarchaeota genome, DRTY7 bin_36, may be the first genome of the pSL12 lineage, which has been found in hot springs previously[20–22].

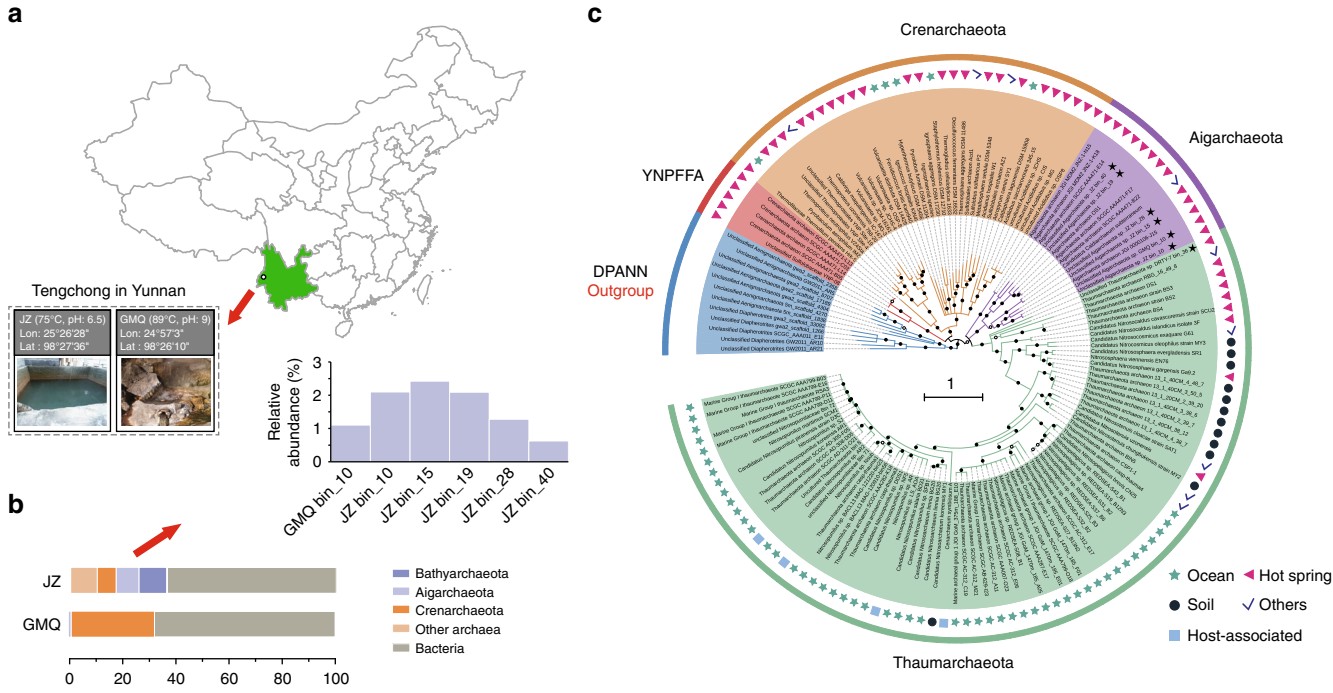

**Fig. 1** The reconstructed genomes of Aigarchaeota. **a** Geographic locations of the two hot spring sites, JZ and GMQ, where sediment samples were collected. **b** Relative abundance of the six aigarchaeal genome bins recovered from metagenomes. **c** Phylogenetic placement of the bins. The tree was constructed based on the concatenated alignment of 16 ribosomal proteins. Nodes with bootstrap value ≥90 (70) are indicated as solid (hollow) circles

**Table 1 General genomic features of the Aigarchaeota and Thaumarchaeota bins reconstructed from the metagenome assembly**

| Bins | Aigarchaeota | | | | | | Thaumarchaeota |
|---|---|---|---|---|---|---|---|
| | GMQ bins_10 | JZ bins_10 | JZ bins_15 | JZ bins_19 | JZ bins_28 | JZ bins_40 | DRTY7 bin_36 |
| No. of scaffolds | 10 | 79 | 25 | 54 | 17 | 25 | 167 |
| Genome size (bp) | 1,230,238 | 1,086,093 | 1,473,345 | 1,654,953 | 1,440,436 | 1,471,612 | 1,241,443 |
| GC content (%) | 53.7 | 55.4 | 37.48 | 62.32 | 51.83 | 51.92 | 36.12 |
| N50 value (bp) | 239,895 | 18,760 | 82,890 | 59,043 | 161,320 | 136,575 | 10,470 |
| No. of protein coding genes | 1,398 | 1,222 | 1,524 | 1,734 | 1,581 | 1,566 | 1,443 |
| Coding density (%) | 90.0 | 92.5 | 90.0 | 89.9 | 94.7 | 92.7 | 91.2 |
| No. of rRNAs | 3 | 4 | 4 | 5 | 2 | 4 | 3 |
| No. of tRNAs | 44 | 36 | 36 | 38 | 41 | 45 | 46 |
| No. of genes annotated by COG[a] | 904(64.6%) | 859(70.3%) | 1099(72.1%) | 1097(63.3%) | 1080(68.3%) | 1055(67.4%) | 757(50.7%) |
| No. of genes annotated by KOG[a] | 341(24.4%) | 323(26.4%) | 398(26.1%) | 396(22.8%) | 405(25.6%) | 385(24.6%) | 283(18.9%) |
| No. of genes annotated by KO[a] | 712(50.9%) | 686(56.1%) | 898(58.9%) | 870(50.2%) | 913(57.7%) | 835(53.3%) | 623(41.7%) |
| No. of genes annotated by InterPro[a] | 710(50.8%) | 669(54.7%) | 847(55.6%) | 904(52.1%) | 849(53.7%) | 848(54.1%) | 664(44.4%) |
| No. of genes annotated by MetaCyc[a] | 338(24.2%) | 327(26.8%) | 432(28.3%) | 436(25.1%) | 442(28.0%) | 388(24.8%) | 276(18.5%) |
| Completeness (%)[b] | 98.06 | 97.09 | 99.03 | 97.09 | 97.57 | 98.06 | 93.69 |
| Contamination (%)[b] | 0 | 0 | 0 | 0 | 0 | 0 | 0.97 |

[a]Functional annotation for the seven genomes was conducted by uploading to IMG database
[b]Genome completeness and contamination were estimated using CheckM (ref. [10])

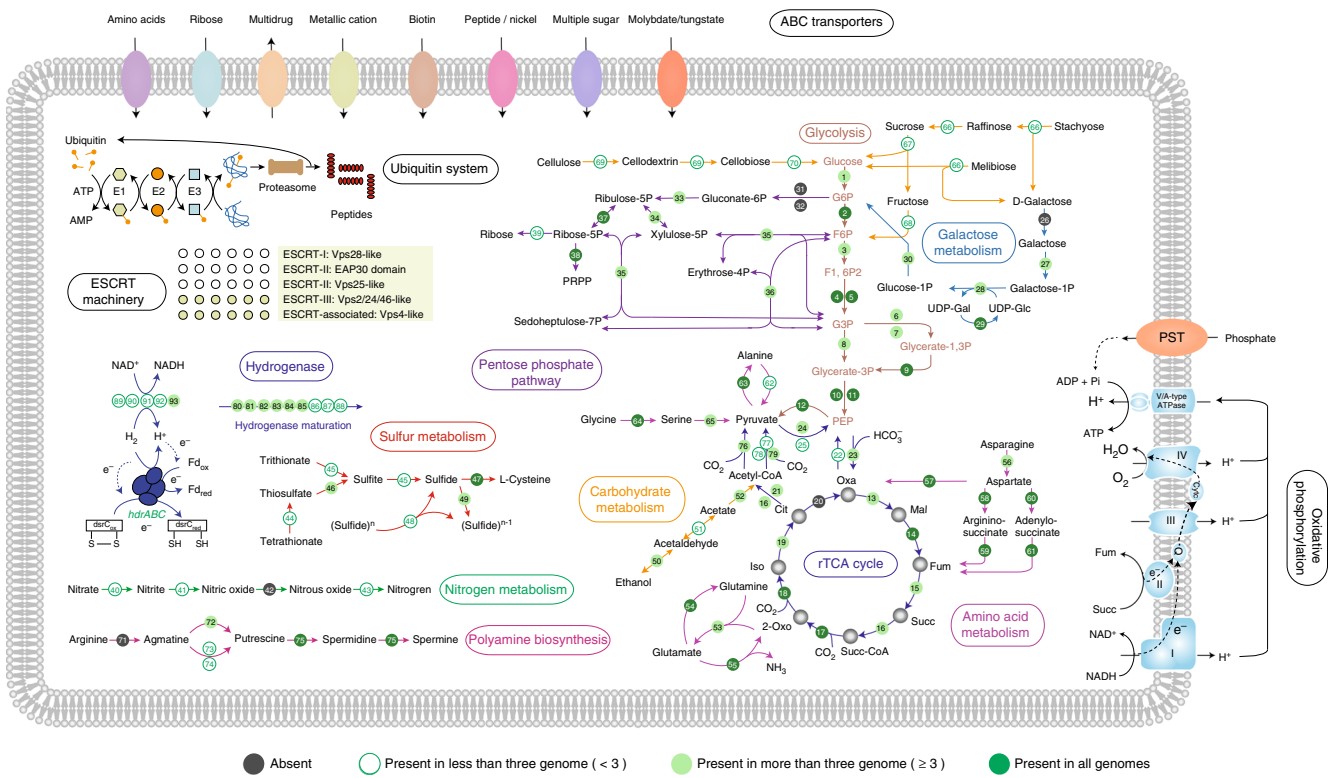

**Fig. 2** Overview of metabolic potentials in Aigarchaeota. Genes related to glycolysis, galactose metabolism, roTCA cycle, the pentose phosphate pathway, the pyruvate metabolism, beta-oxidation of fatty acids, oxidative phosphorylation chain, protein biosynthesis-related pathway, membrane transporters, and ubiquitin system are shown. Green solid circles represent genes present in all six aigarchaeal genome bins. Green hollow circles represent genes occurring in one to five bins. Gray solid circles represent genes absent in all bins. For full names and copy numbers of the genes in number see Supplementary Data 1. roTCA reversed oxidative tricarboxylic acid cycle; ESCRT the endosomal sorting complexes required for transport, G6p glucose 6-phosphate, F6P fructose 6-phosphate; F1,6P$_2$ fructose 1,6-bisphosphate, PEP phosphoenolpyruvate

**Metabolic potential.** Most Aigarchaeota genomes contain complete glycolysis pathways and citrate (TCA) cycles (Fig. 2). JZ bin_40 might also degrade cellulose, as it contains endoglucanase and beta-glucosidase. These genes may enable hydrolysis of cellulose to cellobiose and finally to glucose (Fig. 2; Supplementary Data 1). Amino acid sequence analysis showed that the predicted endoglucanase in JZ bin_40 contained a Frv domain, an aminopeptidase belonging to the M42 family. The identified putative endoglucanases share 33~56% identity to currently known endoglucanases, sharin the same structure including the active sites and potential metal-binding sites (Supplementary Fig. 5). Despite the difficulty in distinguishing them based on sequence similarity, previous cellulase assays showed that this might be a new enzyme with hydrolytic activity against cellulose

in thermal ecosystems[23,24]. Additionally, based on the comparison to Carbohydrate-active enzymes (CAZy) database, numerous genes (29) distributed in 15 glycosyl hydrolase (GH) families were observed in JZ bin_40, including genes for alpha-glucosidase (GH31, degradation of starch and disaccharides), beta-glucosidase (disaccharides degradation), alpha-galactosidase (GH36, lactose degradation), alpha-galactosidase (GH4, melibiose degradation), beta-galactosidase (GH1, cellulose or hemicellulose degradation), alpha-mannosidase (GH38, mannose degradation), alpha-amylase (GH57, starch degradation), alpha-L-rhamnosidase (GH78, pectin degradation), and also a glycogen-debranching enzyme (GH63, release of glucose through the breakdown of glycogen) (Fig. 2; Supplementary Table 2). The generated glucose could further support cell growth through oxidation in glycolysis and the TCA cycle. The first-pass KEGG annotation annotated no gene as beta-glucosidase. However, the best hit for one of the genes in JZ bin_40 was identified as beta-glucosidase (GBC70340.1; 99% in query coverage and 86% in amino-acid identity) by BLASTp to the NCBI-nr database manually, underscoring the need for manual curation of annotation results. Additionally, 26 genes involved in oligosaccharide transporters were identified in JZ bin_40 which could be employed to transport mono/polysaccharides into the cell, mainly including 17 and 5 genes involved in multiple and simple sugar transport system, and a complete raffinose/stachyose/melibiose transport system (three genes; Supplementary Data 1).

Both ATP citrate lyase and citryl-CoA synthase/citryl-CoA lyase were not annotated, indicating the lack of carbon fixation by conventional reductive TCA (rTCA) cycle. Instead, an alternative pathway to generate oxaloacetate, encoding citrate synthase, exists across the Aigarchaeota, suggesting the reversed oxidative TCA (roTCA) pathway for carbon fixation, as proposed recently[25]. In another recent study, strong evidence was presented that the enzymatic activity for citrate synthase was detectable under chemolithoautotrophic conditions in *Thermosulfidibacter takaii* ABI70S6, suggesting this reversible enzyme may convert citrate into oxaloacetate and acetyl-CoA[26]. Previous studies suggested that Aigarchaeota had the potential to fix carbon through the Calvin-Benson-Bassham (CBB) and 3-hydroxypropionate/4-hydroxybutyrate (HP/HB) (dicarboxylate/4-hydroxybuyrate (DC/HB)) cycles[3,8]. However, no enzymes encoding RubisCO and 4-hydroxybutyryl-CoA dehydratase were observed, suggesting these two pathways are missing from these Aigarchaeota genomes, consistent with a previous study[27]. Several heme-copper terminal oxidases were identified in all Aigarchaeota except GMQ bin_10 and JZ bin_10, indicating that most of them are aerobes[28] (Supplementary Data 1). GMQ bin_10 and JZ bin_10 might be anaerobes because that they do not possess any clearly recognizable terminal oxidases. Despite the anaerobic lifestyle of these two organisms, near-complete complex I of respiratory chains were observed, including nearly all subunits of NADH ubiquinone oxidoreductase (*nuo*), but with the absence of *nuoK* and *nuoN* (Supplementary Fig. 6a, b). Instead, the detected *mrpC* and *mrpD* may function as alternatives due to their homology to *nuoK* and *nuoN*[29]. Interestingly, the CXXC motif was identifiable in the *nuoD* protein in GMQ bin_10 and JZ bin_10, implicating it may function as a Group 4 NiFe hydrogenases (Supplementary Fig. 6c). The absence of acetaldehyde dehydrogenase and acetyl-CoA synthetase (ADP-forming) suggests that these thermophiles cannot generate alcohol, acetate or butyrate through fermentation.

Recent studies suggested that microbes may have exploited carbon monoxide as a supplemental energy source for their growth[3,7,30]. Pathways for CO metabolism occur frequently among aerobic bacteria[31]. Most Aigarchaeota bins described here contain at least two copies of all three subunits of the *coxLMS*

complex. However, the absence of *cox* genes reinforces the anaerobic characteristics of JZ bin_10 and GMQ bin_10. In total, 13 *coxL* genes were detected among the genomes. Phylogenetic analysis showed that most of them are classified as Form II, a putative CODH, with typical (AYRGAGR) or atypical motifs (PYRGAGR) observed (Supplementary Fig. 7). Only one *coxL* belonging to JZ bin_19 was identified as Form I with an AYXCSFR signature. This suggests Aigarchaeota could oxidize CO aerobically as an energy supplement when organic substrates are limited. We speculate that the supplied energy might contribute to biomass by coupling carbon fixation through the roTCA cycle.

Sulfur and hydrogen are thought to be crucial in energy cycling in thermal habitats[32]. It is noteworthy that all bins appear to encode capacity for sulfur utilization. Dissimilatory sulfite reductase encoded by *dsrA* and *dsrB* were detected in JZ bin_15, suggesting that this bin could perform dissimilatory sulfite reduction. Aigarchaeota *dsrA* and *dsrB* genes are located near the tips of branches with genes from the phylum Firmicutes (Supplementary Fig. 8a), suggesting that JZ bin_15 might have gained this feature by horizontal gene transfer (HGT) from Firmicutes. High amino acid similarity (66 and 63%, respectively) to *Thermanaeromonas toyohensis* ToBE, within the Firmicutes, supports this observation. Consensus coverage depth shows that the sequence assembly of the *dsrAB*-containing scaffold is reliable (Supplementary Fig. 8b). Previous studies also indicate that thermophiles have undergone frequent HGTs, facilitating their adaptation to the harsh environments[33–35]. The genomic bins GMQ bin_10 and JZ bin_10 harbor genes related to sulfohydrogenase but lack beta and gamma subunits, suggesting a deficiency in sulfur reduction[36]. The alpha subunit may function as a Ni/Fe hydrogenase in these two organisms, which may be important for redox balance[37]. Moreover, the gene encoding for sulfide-quinone reductase (*sqr*) was present in all genome bins except GMQ bin_10 and JZ bin_10. This indicates that most Aigarchaeota are chemolithotrophs, harvesting energy from the oxidation of sulfide to elemental sulfur. In most cases, the oxidation of sulfide by *sqr* or reduction of sulfite by *dsrAB* takes place under anaerobic conditions, implicating a facultatively anaerobic lifestyle of these Aigarchaeota. Interestingly, we found several copies of a heterodisulfide reductase (*hdr*) complex in JZ bin_15 (Supplementary Fig. 9). In particular, one copy clustered with subunits of F420-non-reducing hydrogenase (*mvh* complex), was also found in some sulfate-reducing microorganisms[38,39]. The $H_2$ is oxidized by *mvhADG* to generate protons along with electrons. Due to the absence of methanogensis pathways in this bin, we propose that the released electrons might be transferred to HdrABC to reduce $Fd_{ox}$ and DsrC instead of heterodisulfide. Then the reduced DsrC functions as an electron donor by coupling with DsrAB sulfite reductase to perform sulfite reduction[39]. It was also speculated that the detected Mvh: Hdr complex in *Archaeoglobus profundus* might be employed to reduce an electron carrier, which in turn functions as an electron donor of the enzymes of sulfite reduction[38].

In Aigarchaeota, 19 NiFe hydrogenases and 33 hydrogenase maturation factors are present (Supplementary Data 1), potentially involved redox homeostasis[4,36]. Most of these hydrogenases form a separate lineage, showing that they are highly divergent from previous reported hydrogenases in other microbes (Supplementary Fig. 10). Phylogenetic analysis reveals that 7, 5, 3, and 4 of the 19 NiFe hydrogenases are type 3a, 3b, 3c, and 4d hydrogenases, respectively, which mainly include F420-reducing hydrogenase (*frh*), sulfohydrogenase (*hyd*) and Ni,Fe-hydrogenase III (*ech*) (Supplementary Fig. 10; Supplementary Data 1). Several of them are Aigarchaeota-specific and are distantly related to other hydrogenases. The *mvh* complex

described above represents a type 3c hydrogenases. These hydrogenases could supply intracellular reducing equivalents needed for various redox reactions[40,41].

The presence of Mu-like prophages in JZ bin_28 and bin_40 suggests that Aigarchaeota might be subject to phage infection in hot springs. Prophages impose a fitness burden through DNA insertion in the host genome, but can also provide beneficial functions to their hosts to make them competitive and survive in harsh environments. As in most Archaea, several CRISPR loci consisting of CRISPR arrays, belonging to type I and type III systems, were detected in all the genomes (Supplementary Fig. 11). This illustrates that even under the extreme high temperature, microbes in the community could also be invaded by viruses. This is consistent with previous research showing that these two CRISPR-cas systems are ubiquitous among other Archaea[42].

Current available Aigarchaeota genomes were all from hot spring ecosystems, with the exception of "*Candidatus* Caldiarch-aeum subterraneum", which was from hot fracture water within a subsurface gold mine (Supplementary Data 2)[7]. Most of them harbor heme-copper oxidases, showing they are aerobes that can use oxygen as an electron acceptor[3,30]. GMQ bin_10 and JZ bin_10 are anaerobes due to the lack of terminal oxidases. Their phylogenetic position near the tips of the Aigarchaeota illustrates a possible shift from a facultatively anaerobic to a strictly anaerobic lifestyle. In this study, these two bins are divergent from the other four and show dramatic metabolic differences, suggesting that Aigarchaeota is experiencing sympatric evolutionary divergence. Unlike the considerable metabolic diversity within most Aigarchaeota genomes, GMQ bin_10 and JZ_10 are devoid of pathways for carbon monoxide oxidation, aerobic respiration, dissimilatory sulfite reduction, and sulfide oxidation. Carbon fixation is a common trait among most Aigarchaeota genomes by employing either the roTCA cycle (the four genomes in this study) or the 3-hydroxypropionate/4-hydroxybutyrate pathway to fix carbon dioxide[8]. However, the two anaerobes lack this capability due to the absence of key enzymes. Mostly, rTCA or roTCA cycle are only found in anaerobic organisms, however, some studies show that at least rTCA is functional in microaerobic or aerobic microbes such as "*Candidatus* Nitrospira defluvii" and *Leptospirillum* genomes[43,44]. In "*Candidatus* Nitrospira defluvii" genome, near complete complex I-V pathways were identified, indicating the aerobic lifestyle of this microbe. Comparative genomics shows that several Aigarchaeota genomes including "*Candidatus* Caldiarchaeum subterraneum", "Aigarchaeota archaeon OS1", and the four genomes in this study have the ability to employ *sqr* to oxidize reduced sulfide as an electron donor for chemolithotrophy. Other Aigarchaeota lack this capability due to the missing of *sqr* gene. Closely related neighbors JZ bin_15 and Aigarchaeota archaeon JGI 0000106-J15 were the only two genomes possessing *dsrAB* genes with high sequence identity (>90%), indicating they inherited sulfite reduction ability vertically, following earlier HGT transfer from Firmicutes.

**Eukaryotic signatures in Aigarchaeota genomes**. Several eukaryotic signature proteins (ESPs) were identified in Aigarch-aeota genomes as described before[7] (Supplementary Fig. 12; Supplementary Table 3). All the six genomes harbor an ubiquitin modifier system consisting of Ub-like, E1-like, E2-like, E3-like, and Dub-like proteins, which could be used to degrade or recycle damaged proteins[45] (Fig. 2; Supplementary Fig. 13). The whole reaction might be quite similar to the canonical eukaryotic ubi-quitylation process. First, pro-ubiquitin could be cleaved by the Rpn11 metalloprotease homolog (IPR000555), leading to

the exposure of the di-glycine motif of the modifier at the C-terminus. E1-like enzyme (IPR000594) could subsequently be used to adenylate the residue of the di-glycine motif. The activated modifier could then be transferred to the catalytic cysteine of the E1-like enzyme, forming a covalent thioester intermediate. Next, the transfer of ubiquitin from E1 to the active site cysteine of the E2 could then be triggered by the E2-like conjugating enzyme (IPR000608) via a trans-thioesterification reaction. Finally, E3-like enzymes could catalyze the aminolysis-based transfer of the ubiquitin from the E2-like protein onto a specific target protein, which ultimately results in the covalent attachment of a small ubiquitin modifier to a substrate and further degra-dation through the proteasomal degradation pathways[45]. More-over, a total of 64 ribosomal proteins (RP) were observed among the six genome bins. Over half were identified as eukaryotic-like RPs, including RP-S19e and RP-L30e, which were previously only reported to be present in eukaryotes[46] (Supplementary Table 3). RP-L13e and RP-S26e were found in Aigarchaeota, consistent with previous studies suggesting that these two RPs are present in TACK superphylum[47]. In addition, a wide variety of GTPases were identified among Aigarchaeota (Supplementary Data 1). One genomic bin may encode a member of the Rab family of small GTPases, an enzyme thought to predate the origin of Eukaryotes, and generally presumed to be involved in membrane biogenesis by interacting with proteins[48]. Like Crenarchaeota, all genomes encode homologs of ESCRT-III and Vps4 compo-nents[49]. Those genes were proposed to play a pivotal role in cell division, and the inactivation of Vps4 leads to the accumulation of enlarged cells[50]. However, the key genes *cdvB* and *cdvC* are present, but a gene for *cdvA* is absent among all the genomes. Both CdvB and CdvC show low similarity to proteins identified in Eukarya (average ~26% and ~32% identity to the best hits in Eukarya based on the RPS-BLAST). Overall, we found that these archaea are of particular interest because they contain a surprising mixture of bacterial and eukaryotic features. For instance, the sulfite reduction ability in JZ bin_15 is derived from Firmicutes and several genes involved in the roTCA cycle show high amino acid identity to those of the genus *Thermus*. From Bacteria, they can acquire the abilities related to central meta-bolism through HGT. Otherwise, they share common ancestry and mechanistic similarities to the functions related to cell division, DNA replication, and transcription in Eukaryotes.

**The evolutionary history of Aigarchaeota**. The Aigarchaeota represents an evolutionarily diverse group of microbes, which are mainly found in high temperature environments including geothermal springs, the deep subsurface, marine sediments, and marine hydrothermal vents[5,6,8,9,51–53]. It is not yet confirmed whether Aigarchaeota represents a new phylum or represents a deeply branch within the Thaumarchaeota[16]. To better under-stand the relationship between the two groups, we carefully selected 94 genomes including 14 Aigarchaeota and 80 Thau-marchaeota from public databases including IMG-M and NCBI (see Methods for the selection criteria), and built phylogenetic and phylogenomic trees (Supplementary Fig. 14; Supplementary Data 2). A 16S rRNA-based phylogenetic tree places Aigarchaeota as the nearest neighbor to Crenarchaeota (Supplementary Fig. 2), consistent with a previous study[54], suggesting that Aigarchaeota may represent a new phylum compared to Thaumarchaeota. The phylogenomic tree based on the concatenated alignment of 16 ribosomal proteins shows that Aigarchaeota and Thaumarchaeota come from divergent groups with convincing bootstrap support at the parental node of two lineages, providing further compelling evidence that they are different phyla. However, uncertainties are observed for the affiliation of Aigarchaeota archaeon JGI MDM2

JNZ-1-N15 and Aigarchaeota archaeon JGI MDM2 JNZ-1-K18 due to their uncertain relationship to other Aigarchaeota. Revisiting the phylogenetic placement of these two phyla in an RNA polymerase-based tree even put them into Thaumarchaeota with high bootstrap confidence (>70%; Supplementary Fig. 15). However, we still consider them as Aigarchaeota because they group with Aigarchaeota based on the functional similarity based on KO or COG annotation (Supplementary Fig. 16). To our surprise, we found that one lineage comprised of six genomes (hereafter, we refer to them as "transitional genomes") has a closer relationship to Aigarchaeota based on KEGG and eggNOG functional annotations, but both the phylogenetic and phylogenomic trees suggest that they represent members of the Thaumarchaeota (Supplementary Fig. 16). In particular, DRTY7 bin_36, which generated in this study, might be the first genome of deep-branching pSL12 group (Supplementary Fig. 2). Principal coordinates analyses (PCoA) based on COGs (KOs) were conducted to visualize the relationship among different members of the Aigarchaeota and Thaumarchaeota. From the PCoA plot, genomes in these two phyla appear to cluster based on their habitat rather than their taxonomic assignment. One lineage consists mainly of Thaumarchaeota genomes inhabiting cold or ambient temperature environments such as marine or soil. The other cluster is mainly composed of the Aigarchaeota and the "transitional genomes", which are mainly found in high-temperature ecosystems such as terrestrial hot springs (Supplementary Fig. 16). Besides the six transitional genomes, the remaining 71 Thaumarchaeota genomes are all ammonia oxidizers, with six being possible exceptions, possibly because of genome incompleteness (average <62%). Most of them are mesophiles, with only a few from high-temperature

environments. "Candidatus Nitrososphaera gargensis" Ga9.2 was proposed to be the first thermophilic ammonium oxidizer, found in the moderate thermal spring in Russia[14]. The following reported deep-branching AOAs including "Candidatus Nitrosocaldus yellowstonii", "Candidatus Nitrosocaldus islandicus" isolate 3F, and "Candidatus Nitrosocaldus cavascurensis" strain SCU2 show adaptation to even higher temperature[55–57]. Based on this observation, we propose that the last common ancestor of Aigarchaeota and Thaumarchaeota were thermophiles, and the marine Thaumarchaeota are the descendants of the thermophiles. The detected crenarchaeol, a diagnostic membrane lipid, in "Candidatus Nitrosocaldus yellowstonii" also supports the hot-origin of Thaumarchaeota[55]. This evolutionary scenario is consistent with the evolution of the destabilizing cyclohexyl ring of crenarchaeol/thaumarchaeol to allow niche invasion of thermophiles into cold environments. Dispersal into cooler habitats would also trigger evolution of the transitional genomes described here. The later genetic interactions with the community members driven by the environmental selection lead to the loss or acquisition of specific genes from organisms in the new microbial communities (e.g., the derived feature of ammonia oxidation) to make them better adapted to the new habitats (Supplementary Fig. 17).

To decipher the evolutionary histories of the Aigarchaeota and Thaumarchaeota, gene gain and loss events were predicted by mapping the inferred orthology of genes to the Bayesian tree. The Bayesian tree is quite robust with all the nodes showing high posterior probability (Supplementary Fig. 18). The Aigarchaeota and Thaumarchaeota are monophyletic and share over 1000 orthologous genes for their common ancestor (Fig. 3). Several terminal oxidases were gained at node 1 (Supplementary Data 1),

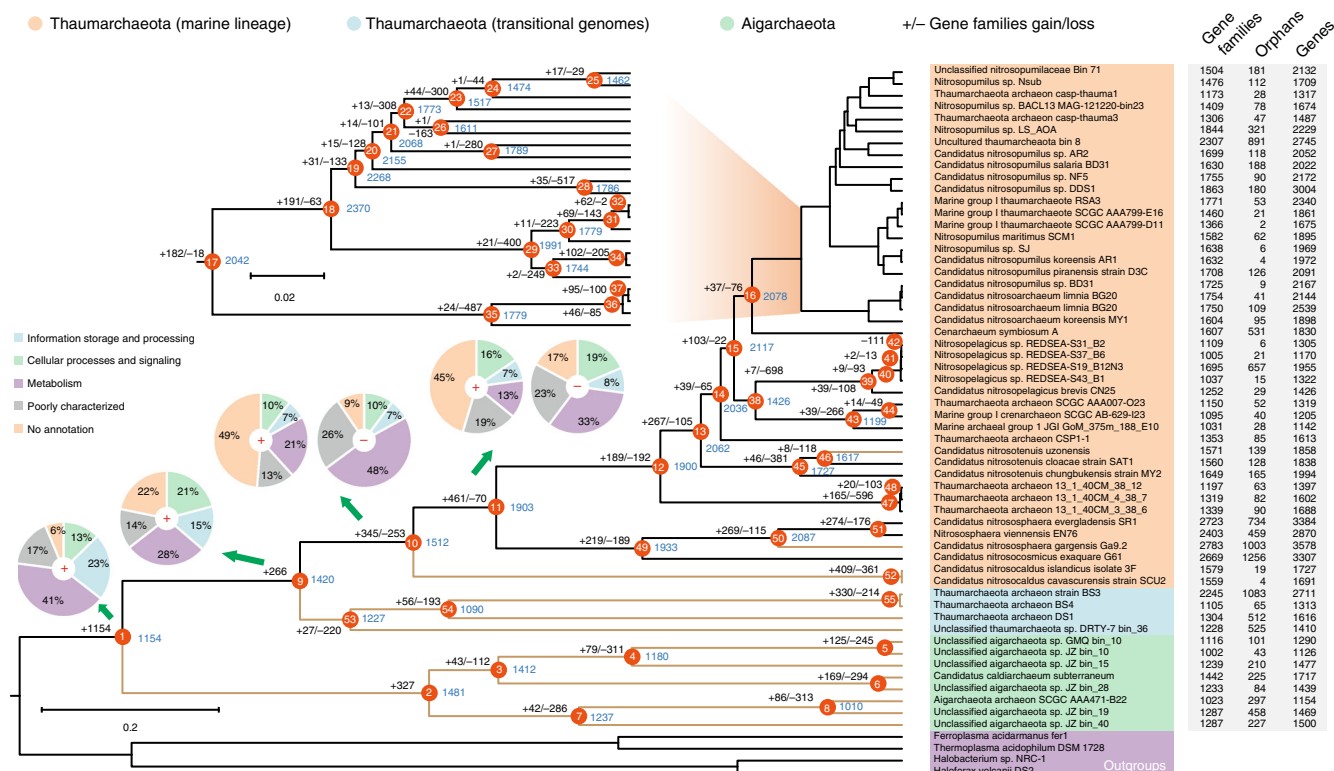

**Fig. 3** Ancestral genome content reconstruction using COUNT software. The tree topology is based on the Bayesian tree generated from MrBayes[78]. The numbers of gain and loss events were marked at each lineage of the tree. "+"s represent gain events and "−"s represent loss events. The red star represents the major gene gain event. The pie chart shows the numbers of gained genes by COG categories. Thermophiles are colored as brown branches in the tree. A list of gained and lost genes for the fours key nodes were shown in Supplementary Data 3

indicating the ancestor of these two lineages were aerobes. Aigarchaeota are reported to be hyperthermophiles or moderate thermophiles, and are widely distributed in different thermal habitats including terrestrial, marine, and subsurface environments[9]. Hence, it appears to be reasonable that the ancestor of the Thaumarchaeota likely also originated from high-temperature habitats because little has been changed as revealed by comparing the lineage represented by transitional genomes and the parental nodes of Aigarchaeota (Fig. 3). The evolutionary scenario was also strengthened by a previous study about the "hot" origin of Thaumarchaeota[58]. Then, we assume that niche invasion from thermal to moderate temperature habitats might have selected for transitional genomes and subsequently canonical Thaumarchaeota. New resources after the colonization of these new habitats might have triggered adaptive radiations, forming the largest branch of the Thaumarchaeota. This finding was further supported by the observation that a large number of gene gain events occurred at nodes 10 and 11 of the tree, taking up ~23 and 24% of gene families at corresponding nodes (Fig. 3). These two steps of gene acquisition largely expanded the genetic diversity of marine Thaumarchaeota, leading to the functional divergence from ancestral thermophilic Thaumarchaeota and Aigarchaeota. Of the total 806 gained genes at nodes 10 and 11 (Supplementary Data 3), further analysis showed that 213 (~62%) and 296 (~64%) of them were identified as hypothetical proteins based on comparison to nogSTRING databases (Fig. 3). In comparison, most of the lost genes at nodes 10 and 11 are related to basic metabolism, including transporters and dehydrogenases (Supplementary Data 3). Additionally, several genes related to nitrogen metabolism including *ureABCDEF* and *amoABC* operons were acquired at node 10, indicating that ammonia oxidation is a derived feature of Thaumarchaeota. The *amoABC* genes in "Candidatus Nitrosocaldus islandicus" 3F and "Candidatus Nitrosocaldus cavascurensis" SCU2 suggest AOAs might originate from thermal habitats, which was also supported by previous findings[7,59,60]. The inherited ammonia oxidation ability could sustain their growth in nutrient-depleted marine and terrestrial environments with low concentrations of ammonia, allowing them to compete for ammonia as an electron donor and providing a selective advantage relative to AOB, heterotrophs, and phytoplankton[61]. Urea assimilation is employed by most AOA to supply sufficient ammonia by catalyzing conversion of the urea molecule to two ammonia molecules and one carbon dioxide molecule. Besides the large genetic investment in nitrogen metabolisms, several genes involved in cobalamin ($B_{12}$) synthesis were observed at node 10, which is necessary to catalyze methyl transfer reactions in amino acid and DNA synthesis[62]. Previous study revealed a common phenomenon that the $B_{12}$ biosynthesis pathway was widely distributed among oceanic Thaumarchaeota genomes[63], serving as a potential source of multiple B vitamins required by other community members[64]. However, based on our observation, they might be also inherited from thermophilic AOAs (Fig. 3).

We estimated that the ancestor of the Aigarchaeota possessed more gene families than the extant organisms. As time went on, gene loss events occurred frequently in all lineages. To gain a better understanding of this process, we investigated variants of the six genomes to obtain detailed insight into the influence of environmental selection on genomic evolution. Single nucleotide polymorphism (SNP) calling was conducted as described in Materials and Methods. The detected SNPs in the six genomes are comparable except JZ bin_10, which has the smallest genome size, and the most SNPs (23321; Supplementary Table 4). In contrast, GMQ bin_10, the closest neighbor to JZ bin_10 (97% AAI), has a larger genome but fewer SNPs (2386; Supplementary Table 4). More than half of the SNPs in

almost all the genomes are synonymous mutations, illustrating that in most cases, those mutations had no effect on their growth in extreme thermal habitats. Almost all the $dN/dS$ values were <1, indicating that purifying selection was acting to remove deleterious mutations.

**Genome expansion through HGT.** HGT is in important evolutionary process in prokaryotes that has great impact on the diversity of gene repertoires, especially for those microbes in extreme habitats[35,65]. Putative HGT events have contributed substantially to genome contents of Aigarchaeota (Supplementary Table 4). The transferred genes comprise predominantly basic metabolic functions, with amino acid transport and metabolism (~18.5% of the total HGTs), energy production and conversion (~15.6%), carbohydrate transport and metabolism (~9%), nucleotide transport and metabolism (~5.5%), lipid transport and metabolism (~4.8%), inorganic ion transport and metabolism (~6.7%), secondary metabolites biosynthesis, transport, and catabolism (~2.9%), and coenzyme transport and metabolism (~6.1%) being significantly enriched in this eight functional classifications (Two-tailed Fisher's exact test with confidence intervals at 99%, $P < 0.05$; $P$ values were adjusted with the "BH" criteria) (Supplementary Fig. 19). As is typical, informational proteins (e.g., ribosomal proteins, DNA polymerases) underwent fewer HGTs than other gene families (Supplementary Fig. 19). Among the identified potential HGTs, most appear to be acquired from the same domain, including Crenarchaeota and Euryarchaeota (Fig. 4). Bacteria also contributed substantially to generating genetic diversity through HGT, and several genes were transferred from Firmicutes and Proteobacteria (Fig. 4). These results might seem inconsistent with previous studies showing that closely related organisms engage in more frequent genetic exchange than distantly related ones[66]. However, interdomain gene transfer is highly asymmetric in that Archaea acquire genes much easier from bacteria than vice versa[67]. Gene acquisition from bacteria appears to have provided the key innovations for Aigarchaeota, facilitating their access to new niches. For example, several CODHs and hydrogenases were identified as HGTs with bacteria serving as donors (Supplementary Data 4). As reported, carbon monoxide and hydrogen are ubiquitous and are believed to be primary substrates and energy provider in hydrothermal habitats[68]. For most members of Aigarchaeota in this study, they are able to grow via the oxidation of carbon monoxide coupling with oxygen reduction. Several genes related to oxidative phosphorylation in GMQ bin_10 and JZ bin_10 were found in large gene clusters, indicating they were likely acquired in a single event (Supplementary Data 4). The associated genes may be donated by Firmicutes through HGT. This finding provides important clues about the origin of aerobic Aigarchaeota. Firmicutes also endowed JZ bin_15 with sulfite reducing capability with strong evidence as described above (Supplementary Fig. 8a). The roTCA cycle might be a derived feature for most bins except GMQ bin_10 and JZ bin_10. The key enzyme citrate synthase was identified to be imported from Euryarchaeota or Deinococcus–Thermus (Supplementary Data 4). Hence, HGT might be the main driver to provide genetic variability that allows different types of Aigarchaeota to occupy distinct niches, resulting in the functional partitioning and ecological divergence within the same habitat.

## Conclusions
Aigarchaeota poorly understood but important lineage that is abundant and widely distributed in thermal habitats. The high-quality genomes described here are diverse both in metabolic pathways and ecological roles, suggesting functional partitioning

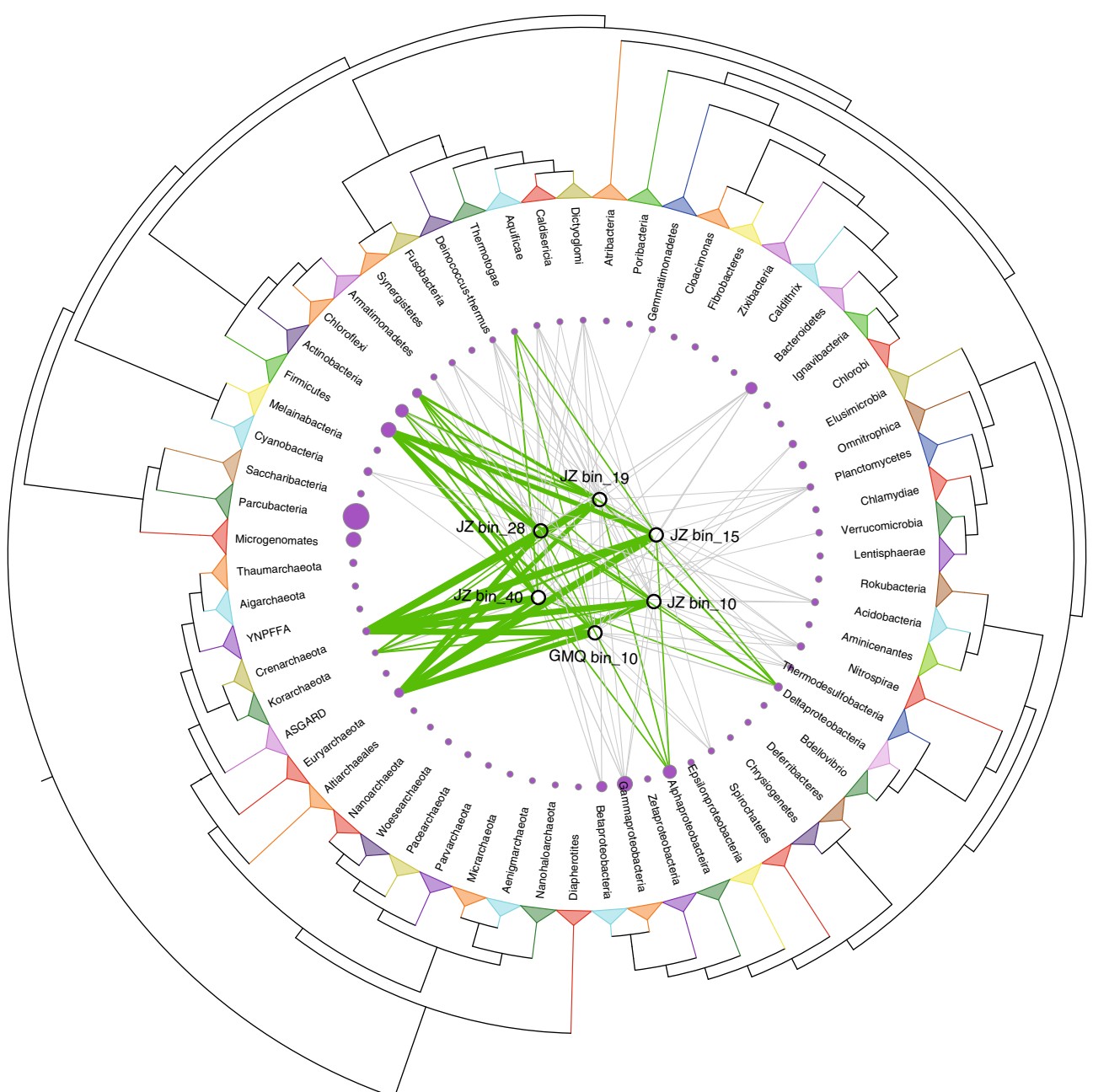

**Fig. 4** Extensive horizontal gene transfer events detected in Aigarchaeota. The maximum likelihood tree was pruned and redrawn at the phylum level according to the tree of life published by Hug et al.[79]. Green edges represent potential gene flows from the corresponding phylum involving ≥10 genes, and gray lines are the ones with <10 genes. Edge thickness is proportional to the number of putatively transferred genes. The sizes of red circles are proportional to the number of reference genomes collected for that phylum

and ecological divergence within a single geothermal region. To avoid competition among closely related microbes, HGT contributes a lot to their genome innovations and allows them to utilize different kind of substrates. Intriguingly, several of the acquisitions are imported from Bacteria, indicating inter-domain genetic interactions play a key role in shaping the genetic diversity of Aigarchaeota. From the evolutionary perspective, our research supports the hypothesis that the last common ancestor of Aigarchaeota and Thaumarchaeota have a large genome size and both of them are originated from thermal habitats. Overall, this discovery is expected to have a substantial impact on our understanding of their roles in biogeochemical cycling and the evolutionary history of the important but poorly understood phylum Aigarchaeota.

## Methods

**Sample acquisition, DNA extraction, and metagenomic sequencing**. Two hot spring sediment samples including JZ and GMQ were obtained from Tengchong, Yunnan, China. Detailed site description as following: (1) *GMQ*: A small source pool with a high flow rate (10.4 L/S), with length, width and depth around 98, 79, and 9.5 cm, respectively. Spring water is clear, and silicate sand sediment could be seen at bottom. The spring is surrounded by bush and grass. Leaf litter and other debris were observed in this spring. (2) *JZ*: Man-made cubic well with side length 400 cm, and depth 150 cm, covered with a ceiling. Water is not clear, bottom is not visible. Mineral deposition was observed on the well walls. The GPS coordinates of the locations from which samples were listed in Fig. 1. Both samples were collected using sterile spatulas and spoons and stored in liquid nitrogen before transporting to the lab. Community genomic DNA was extracted from approximately 20 g of sediment material using PowerSoil DNA Isolation kit (MoBio). DNA concentrations of the extract and constructed libraries (with insert size of 350 bp) were measured with a Qubit fluorometer. Metagenomic sequence data for the two samples are generated using Illumina Hiseq 4000 instruments at Beijing Novogene

Bioinformatics Technology Co., Ltd (Beijing, China). The amount of raw sequence data was ~30 Gbp (2 × 150 bp) for each sample.

**Metagenome assembly and genome binning**. Preprocessing of raw data was carried out as described previously[69]. The quality metagenomic sequences for each sample were de novo assembled separately using SPAdes (version 3.9.0)[70] with the following parameters: −k 33,55,77,99,111-meta. Then, GapCloser (version 1.12; http://soap.genomics.org.cn/) was used with default parameters to fill a proportion of gaps of the assembled scaffolds. Genome binning of the metagenomic assemblies was conducted with a combination of emergent self-organizing maps (ESOM)[71] and MetaBAT[72]. In brief, scaffolds with length <2500 bp in each assembly were removed from the further analysis. Qualified metagenomic data from samples was mapped to each assembly to compute the coverage information using BBMap (version 38.85; http://sourceforge.net/projects/bbmap/) with the parameter as: $k = 15$ minid = 0.9 build = 1. Then coverage information of the scaffolds and tetra-nucleotide frequency (TNF) were used to perform the genome binning which conducted by MetaBAT[72]. All bins were subjected to manual examination to remove contaminations. Specifically, genome bins were sheared into short fragments (5 to 10 kb) and clustered based on the TNF using ESOM[71]. Then the generated clusters were visualized to judge the situations that if one genome bin has been split into sub-blocks or sub-blocks could be merged into one single bin (Supplementary Fig. 1). To improve the accuracy of genome bins, the obvious discordant points were deleted manually from the clusters. The completeness, contamination and strain heterogeneity of each bin were evaluated using CheckM (version 1.0.5)[10]. To get the optimal quality of genome bins, clean reads for each genome bin were recruited using BBMap with the parameters listed above. Then, genome bins were reassembled by SPAdes (version 3.9.0)[70] with the following parameters: --careful -k 21,33,55,77,99,127. Finally, six genome bins belonging to Aigarchaeota were retained for the further analysis. The subsequent functional annotation, phylogenetic and phylogenomic inference and metabolic pathway reconstruction were conducted as detailed described in the Supplementary Notes 1, 2.

**Comparative genomics**. All genomes classified under Aigarchaeota or Thaumarchaeota were downloaded from NCBI (https://www.ncbi.nlm.nih.gov/) and IMG-M (https://img.jgi.doe.gov/cgi-bin/m/main.cgi) databases. CheckM[10] was used to check the genome quality. Genomes with redundancy and completeness < 50% were removed for the further analysis. Finally, a total of 94 draft genomes were taken into comparison. Seven were from our study including six and one belonged to Aigarchaeota and Thaumarchaeota, respectively. The remaining 87 genomes from the public databases include eight aigarchaeal and 79 thaumarchaeal genomes (Supplementary Data 2). To compute the amino acid identity (AAI) of each pair of genomes, orthologous genes were identified based on reciprocal best BLAST hits ($E$-value < 1e$^{-5}$) based on their predicted amino acid sequences. AAI was calculated as the mean similarity of all orthologous genes.

For the further comparative analysis, only genomes with completeness >80% were taken into consideration, respectively, which lead to the filtration of extra 38 genomes (Nitrosopumilus sp. AR was also removed due to the high contamination estimated by CheckM). Clusters of homologous protein were reconstructed for the remaining 56 genomes. An all-against-all genomes BLAST were conducted using the thresholds $E$-value <1e$^{-10}$ and sequence identity ≥30%. MCL (−I 1.4)[73] was used to reconstruct protein clusters based on the reciprocal best BLAST hits (rBBHs). This yielded a total of 18,894 protein families with 12,198 were classified as singletons. To address the evolution of life histories of the two phyla, ancestral family sizes were inferred using the program COUNT[74] with Dollo parsimony. This approach strictly prohibits multiple gains of genes and allows reconstructing gene gain and loss events at both observed species and potential ancestors (leaves and nodes on the phylogenetic tree).

Putative HGTs were inferred using HGTector[75]. Homologs of predicted genes were retrieved from the NCBI-nr database using BLASTp as implemented in NCBI BLAST +2.2.28. Quality cutoffs for valid hits were $E$-value ≤1e$^{-20}$, sequence identity ≥30%, and coverage of query sequence ≥50%. Putatively HGT-derived genes were defined as those with hits from within the Thaumarchaeota (NCBI TaxID 651137, which includes Aigarchaeota) significantly underweighted, and with hits outside the phylum not underweighted.

**SNP detection and *dN/dS* calculation**. The quality reads were mapped to the corresponding genomes using BBMap (minid = 0.97). GATK[76] was then used to generate the primary high-quality variant calls including SNPs and InDels. Custom Perl scripts were developed to identify and classify all the SNPs as intergenic, intragenic, synonymous ($dS$) or non-synonymous ($dN$). Calculation of $dN$ and $dS$ values between GMQ bin_10 and JZ bin_10 were performed using the yn00 program in the PAML package (version 4.9)[77]. Both the Nei-Gojobori and Yang and Nielsen methods were used. To remove the bias caused by too few $dS$, genes with $dN/dS$ ratio higher than 5 were excluded from further analysis.

**Data availability**. The genome bins described in this study have been deposited at JGI IMG-MER under the Study ID Gs0127627 and WGS accessions Ga0180324 (Unclassified Aigarchaeota GMQ bin_10), Ga0180307 (Unclassified Aigarchaeota

JZ bin_10), Ga0180308 (Unclassified Aigarchaeota JZ bin_15), Ga0177927 (Unclassified Aigarchaeota JZ bin_19), Ga0180309 (Unclassified Aigarchaeota JZ bin_28), Ga0180310 (Unclassified Aigarchaeota JZ bin_40) and Ga0181444 (Unclassified Thaumarchaeota DRTY7 bin_36). The datasets generated during and/or analyzed during the current study are available from the corresponding author on reasonable request.

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

## Acknowledgements

We thank the Guangdong Magigene Biotechnology Co., Ltd. China for the assistance in data analysis, and the entire staff from Yunnan Tengchong Volcano and Spa Tourist Attraction Development Corporation for strong support. This work was financially supported by the Key Project of International Cooperation of China Ministry of Science and Technology (No. 2013DFA31980), Science and Technology Infrastructure work project of China Ministry of Science and Technology (No. 2015FY110100), the National Natural Science Foundation of China (31600298, U1201233, 31470139 and 31370154), Natural Science Foundation of Guangdong Province, China (No. 2016A030312003), the China Postdoctoral Science Foundation (2016M602567), the Guangdong Province Key Laboratory of Computational Science and the Guangdong Province Computational Science Innovative Research Team. W.J.L. was also supported by Guangdong Province Higher Vocational Colleges and Schools Pearl River Scholar Funded Scheme (2014).

## Author contributions

Z.S.H., Y.N.Q., R.K. W.S.S., and W.J.L. conceived the study. L.L. performed the measurement of physiochemical parameters. Y.R.Y. and Y.T. performed the DNA extraction. Z.S.H. Y.N.Q., Y.X.L., Y.Z.R, C.J.C., E.M.Z., and Y.L.Q. performed the metagenomic analysis, genome binning, and functional annotation. Z.S.H. and Q.Z. performed the detection of horizontal transferred genes and related evolutionary analysis. Z.S.H., Y.N.Q., W.J.L., R.K., B.P.H., and Q.Z. wrote the manuscript. All authors discussed the results and commented on the manuscript.

## Additional information

**Competing interests:** The authors declare no competing interests.

