## [Peer Review File · Nature Communications]

Reviewers' comments:

Reviewer #1 (Remarks to the Author):

The paper by Hua et al, entitled "Genomic inference of the metabolism and evolution of the archaeal phylum Aigararchaeota in hot springs" provides the scientific community with six new high-quality metagenomes belonging to the poorly characterized Aigararchaeota phylum. By using a combination of evolutionary (meta)genomic approaches, the authors reconstruct Aigararchaeota metabolism, including ESP distribution and conclude that Aigararchaeota are facultative anaerobes, and that most are chemolithotrophs that can perform sulfur oxidation. They also analysed the phylogenetic relationship of this phylum with thaumarchaeota, propose an evolutive scenario considering a hot origin of these two phyla whose ancestor shared 1085 genes, detect intra- and inter domain HGTs and identify six transitional thaumarchaeota genomes whose KEGG and eggNog functional annotations places them close to Aigararchaeota genomes.

The paper is very well written and structured, high some high quality figures and presents an analysis covering many relevant aspects regarding Aigararchaeota (and Thaumarchaeota evolution). However, after a careful analysis of the supplementary data provided, I feel that some of the authors' claims, specially in what regards Aigararchaeota metabolic abilities and ancestral genomic reconstruction are not fully supported/can not be evaluated by the presented data.

For example:

Line 114-115 (and abstract) –"The pathway responsible for respiration encoded by the nuo operon was identified in all Aigararchaeota, indicating that they are aerobic". The presence of the nuo operon per se is not an indication of aerobic respiration. Looking to the scarce information provided in table S1, two out of the six genomes do not contain complex IV (nor complex II or Complex III). Unless other terminal oxidases are present in the genomes, with the data presented, GMQ bin_10 and JZ bin_10 are anaerobic organisms and not facultative aerobes. With 1/3 of the genomes being anaerobic, Aigararchaeota should not be classified as facultative aerobes.

Also, the absence of some of Complex I subunits made me wonder if these complexes are really part of complex I or might belong to mbh/x or related complexes that catalyze different reactions (see for instance Marreiros et al, 2013 BBA-Bioenergetics). Usually, the subunits of these complex are annotated as Nuo genes or mrp antiporters in KEGG. In fact, the above mentioned lineages seem to be metabolic (very) different from the other 4 genomes also by the absence (as the authors well state) of sqr and CO metabolism and a quick inspection of the genes acquired by HGT, possibly also devoid of heme containing proteins.

Figure 2 as presented is highly misleading, because the presence of a gene in 5 out of 6 or

the presence of a gene in a single organism have very different implications in the interpretation of the overall Aigararchaeota metabolism. If some cases are well explained in the text, others are not so clear.

I also missed a full description of the metabolic reconstruction of each one of the six genomes, with their KEGG/COG etc annotations, involvement in which metabolic pathway as well as protein family membership (present in which of the over 6000 MCL protein families, is this gene a singleton). This would allow a full inference of the metabolism of each of the genomes sequenced as well as a better evaluation of the author's conclusions.

I have some reservations regarding the inclusion of genomes less than 90% complete into an ancestral gene content reconstruction since the Count analysis might lead to incorrect inferences of gene gains and losses. In any case, with exception of the cases discussed in the text and general categories, information is given regarding the 1085 genes shared by both phyla, the ones count attributed to being present, gained or lost in the ancestor of aigarchaeota and/or in the ancestor of thaumarchaeotal lineages. A list of these genes would be much appreciated and would give better insights on the metabolic evolution of the lineages.

Based on large state-of-the art phylogenetic analysis, the "hot" origin of thaumarchaeota (and in fact, of Archaea as domain) was already proposed (Raymman et al PNAS 2015). This could strengthen even further the evolutive scenario in here proposed.

Last, not being too familiar with HGTector, after reading the original paper and the methods in here provided I have a methodological question. From what I understood, the count and HGTector analyses were performed independently. How congruent are they? How many cases have Count attributed the origin (and possibly subsequent loss) of a protein family in the ancestor of a lineage and HGTector identify the same protein family as independent cases of HGTs?

Minor:

1- Although DRTY7 genome appears in several figures (e.g Figure S2, Figure X), and is marked as a genome from this study, no further information (or I missed it) is provided regarding the quality of this draft.

2- Recently, the ubiquitin system of *C. subterraneum* (Aigararchaeota) was functional characterized. This might be a good reference when addressing the presence of the ubiquitin system within these organisms. Hennell James et al, Nat. Comm, 2017

Reviewer #2 (Remarks to the Author):

The authors obtained a total of 6 metagenomic bins belonging to Aigarchaeota from two hot springs in China, and discussed their physiology especially for carbon and energy metabolisms. In addition, evolutionary relationship between Aigarchaeota and Thaumarchaeota was discussed. The genomic information of Aigarchaeota, a member of TACK superphylum, has been limited despite its importance in exploring the origin of eukaryotic cellular system and evolution of the domain Archaea. The findings from the 6 bins will contribute to understand the genomic diversity and evolution of Aigarchaeota. However, there are issues through the manuscript described below.

L46: Why did the authors define "energy-deficient". No data presented about this issue in this manuscript. In energy deficient environment, microbial mat formation is not expected while Aigarchaeota has been detected in microbial mat formation.

L66: Nunoura et al. 2010 is likely the first report detecting aigarchaeal SSU rRNA gene sequence from a deep-sea hydrothermal environment.

L83-: I could not identify the relationship between the metagenomic bins and metagenomic libraries through the text, tables and figures through the manuscript including supplementary materials whereas a total of 5 samples were sequenced (L327). Did the authors co-assemble multiple shotgun libraries?
Bowers et al. 2017 should be referred for evaluating metagenomic bins.

L99-: Takami et al. 2015 should be referred in this paragraph.

L100-: Cellulase is very diverse and its characterization is very complicated. The information described here is not sufficient to characterize their properties especially the one manual curation was required.

L106-: New line is required for the discussion of carbon fixation pathways.

L106: The proposal of a new rTCA cycle described in Goltsman et al. 2009 is interesting, but both enzymatic activity measurements and metabolomics are absent. Thus, the authors should repress the interpretation of the potential rTCA cycle in Aigarchaeota.

L114-115: What does nuo operon mean? Appropriate reference should be provided after aerobic.

L127: Sulfur and hydrogen are important energy source even in aerobic environments.

L134-: "this study" should be deleted. Only one example of the potential HGT is discussed here. The manuscript discussed only aigarchaeal metagenomic bins and no data was presented about HGT in the hot spring environment.

L150: Homologues of heterodisulfide reductase are often identified in genomes of non-methanogenic anaerobes although their functions are unknown in most of the cases. It is very curious to mention "By consuming the CoM-SS-CoB generated by methanogens". Did the authors have any evidence about it? It is not allowed based on the data set presented in this manuscript.

L173-: Hennel et al. 2017 should be referred.

L173-: After the paragraph, other aigarchaeal genomes obtained in other study should be combined because the three paragraphs discussed general genomic features of this phylum.

L177: Hershko and Ciechanover 1998 is the reference for eukaryotic ubiquitin system, and appropriate reference of aigarchaeal ubiquitin system should be presented. In addition, I think Aigarchaeota harbor E3-like protein but not E3 protein.

L185: Delete "other".

L197-: The genome information of Nitrosocaldus (Abby et al. 2018) should be added in the revision.

L205-: Did the authors test concatenated RNA polymerase tree?

L222-: Information of Nitrosocaldales represented by Nitrosocaldus and of pSL12 lineage should be provided. pSL12 lineage (also called 1.1c etc. e.g. Weber et al.) is also important in the physiological evolution of Thaumarchaeota because it probably lacks capability of ammonia oxidation.

L230-: The calculation of the divergence age is interesting but is not important in this manuscript because discussion about the paleoenvironmental information was absent in this manuscript.

L259-: The number of aigarchaeal genomes is not sufficient to discuss a genome streamlining pattern.

Fig. S2: This phylogenetic tree did not refer previous reports in naming of each group. For example, in the case of Marin Group I (also called 1.1a), grouping has been proposed in Massana et al. (2000), Takai et al. (2004) and Lauer et al. (2016). The information about Nitrosocaldus and pSL12 group was also absent as described above. Stiegmeier et al (2014) in The prokaryotes- Other major lineages of Bacteria and the Archaea is also helpful.

Reviewer #3 (Remarks to the Author):

Overall, I find this to be a very interesting and well conducted study that will be a valuable contribution to our understanding of the environmental niche of a poorly understood phylum

of archaea.

I have, however, reservations about how some of the phylogenetic analyses were conducted or interpreted. That part of the ms needs to be revised before the ms is acceptable for publication. In particular, I have problems with the concept of deriving divergence times from phylogenetic trees of genes/organisms for which no cross-validation exists. Such analyses rest on the idea that it is possible to extrapolate validated "hard" fossils (that only date back a few hundred million years) into the Hadean and Archean past (2.5-4 billion years ago). There is no scientific support for this concept. On the contrary, past studies have shown that different genes' and organisms' "molecular clocks" "tick" at different intervals. This observation, together with the fact that the concept that today's organisms/genes/habitats are representative of genes/lifeforms/environments that existed billions of years ago is an untestable hypothesis, makes the extrapolation from today's genes/organisms into the past not only speculative but unscientific. I strongly urge the authors to remove any of this data from their ms, because it weakens an otherwise great study.

In addition, I ask the authors to please provide more environmental metadata that would help us to better interpret some of the findings in the genomes, especially the ones that relate to elemental cycling and interactions with the environment and other organisms. The bare minimum would be a description of the temperature, pH, TOC, DOC, DIC, DON, and most important cat- and anions (trace metals would be good but are not considered a must) in the two springs. Was there grass growing around the springs that could provide a source of cellulose/lignin? Ideally, this data would then be put into context to the environments from which the other 8 published aigarchaeotal genomes have been obtained.

Detailed comments

I suggest to shorten the title to read "Genomic inference of the metabolism and evolution of the archaeal phylum Aigarchaeota"

L223-226 and 237-239, this is not exactly true. Hatzenpichler2008 identified the first thermophilic ammonia oxidizer, a discovery that was soon followed up by the report of *Nitrosocaldus yellowstonii*, which is adapted to much higher temperatures. Hatzenpichler postulated "that archaeal ammonia oxidation evolved under thermophilic conditions with the mesophilic lifestyles exemplified by soil or marine AOA likely representing independent, secondary adaptations to lower temperatures." This idea was later supported by the idea of a thermophilic origin of Thaumarchaeota in general, see for example:

Brochier-Armanet C, Gribaldo S, Forterre P. 2012. Spotlight on the Thaumarchaeota. *ISME J.*;

de la Torre JR, Walker CB, Ingalls AE, Könneke M, Stahl DA. 2008. Cultivation of a thermophilic ammonia oxidizing archaeon synthesizing crenarchaeol. *Environ. Microbiol.*

Groussin M, Gouy M. 2011. Adaptation to environmental temperature is a major determinant of molecular evolutionary rates in archaea. *Mol. Biol. Evol.*
Nunoura T, et al. 2011. Insights into the evolution of Archaea and eukaryotic protein modifier systems revealed by the genome of a novel archaeal group. *Nucleic Acids Res.*

L233-234 and 239-241, timing of evolutionary events cannot be inferred from phylogenetic trees alone but needs external validation; the only study that I am personally aware of that ever achieved such validation was a study on bacterial insect-symbionts. In that case the timing could be cross-validated with insect fossils.

While some studies - including the one by David and Alm - claim to reliably date the timing of major evolutionary events, these are in fact untested (often even untestable) hypotheses. They all rest on the idea that it is possible to extrapolate validated "hard" fossils (that only date back a few hundred million years) into the Hadean and Archean past (3-4 billion years ago). Neither is there support for this idea, nor is there support for the concept that "molecular clocks" indeed "tick" in a steady interval and that they do not differ between different organisms or genes living in different habitats.

The authors also do not discuss why they choose the divergence time of *Sulfolobus/Aeropyrum* and not any other group for calibration. In the end, it doesn't really matter because all these divergence times are mere speculation. The study by David and Alm is flawed, because the data features not only many misalignments but also rests on the (flawed) comparison of paralogues (rather than homologous) sequences/protein families.

Fig3, I suggest to either completely remove the figure from the ms or keep the underlying tree but erase the divergence dates from the tree; besides the fact that I have severe doubts about putting time stamps onto phylogenetic trees, I am surprised that the tree is so well resolved and does not show any multifurcations. Please indicate the Bayesian values for every node, and discuss why you chose not to collapse nodes with low Bayesian support values

FigS2 and S6, bootstrap support >50% is the bare minimum; please indicate only nodes >70 and >90% support; all other nodes should be collapsed because they must be considered unreliable

It is not documented how phylogenetic analyses of *DsrAB*, *cox*, and *nif* genes/proteins were performed and how bootstrapping support values were derived

In terms of the two genes predicted to be involved in cellulose degradation, please discuss if any other genes suggest the involvement of *Aigarchaeota* in cellulose degradation; how would these enzymes be transported outside of the cell and how would mono/di saccharides reach the inside of the cell? are any other archaea known to be involved in cellulose degradation? Only bacteria come to mind, but I do not appreciate the latest literature

Responses to the reviewer's comments

Reviewer #1 (Remarks to the Author):

The paper by Hua et al, entitled "Genomic inference of the metabolism and evolution of the archaeal phylum Aigarchaeota in hot springs" provides the scientific community with six new high-quality metagenomes belonging to the poorly characterized Aigarchaeota phylum. By using a combination of evolutionary (meta)genomic approaches, the authors reconstruct Aigarchaeota metabolism, including ESP distribution and conclude that Aigarchaeota are facultative anaerobes, and that most are chemolithotrophs that can perform sulfur oxidation. They also analyzed the phylogenetic relationship of this phylum with Thaumarchaeota, propose an evolutive scenario considering a hot origin of these two phyla whose ancestor shared 1085 genes, detect intra- and inter domain HGTs and identify six transitional thaumarchaeota genomes whose KEGG and eggNOG functional annotations places them close to Aigarchaeota genomes.

The paper is very well written and structured, hight some high-quality figures and presents an analysis covering many relevant aspects regarding Aigarchaeota (and Thaumarchaeota evolution). However, after a careful analysis of the supplementary data provided, I feel that some of the authors' claims, specially in what regards Aigarchaeota metabolic abilities and ancestral genomic reconstruction are not fully supported/can not be evaluated by the presented data.

For example:

Comment1: Line 114-115 (and abstract) –"The pathway responsible for respiration encoded by the *nuo* operon was identified in all Aigarchaeota, indicating that they are aerobic". The presence of the *nuo* operon per se is not an indication of aerobic respiration. Looking to the scarce information provided in table S1, two out of the six genomes do not contain complex IV (nor complex II or Complex III). Unless other terminal oxidases are present in the genomes, with the data presented, GMQ bin_10 and JZ bin_10 are anaerobic organisms and not facultative aerobes. With 1/3 of the genomes being anaerobic, Aigarchaeota should not be classified as facultative aerobes. Also, the absence of some of Complex I subunits made me wonder if these complexes are really part of complex I or might belong to mbh/x or related complexes that catalyze different reactions (see for instance Marreiros et al, 2013 BBA-Bioenergetics). Usually, the subunits of these complex are annotated as Nuo genes or mrp antiporters in KEGG. In fact, the above-mentioned lineages seem to be metabolic (very) different from the other 4 genomes also by the absence (as the authors well state) of *sqr* and CO metabolism and a quick inspection of the genes acquired by HGT, possibly also devoid of heme containing proteins.

Response: We thank the reviewer for the positive assessment and valuable comments. We agree that the *nuo* operon is not an indicator of aerobic respiration. Instead, terminal oxidases like cytochrome c oxidase could be used to judge whether an organism is an aerobe. We conducted a further scan for terminal oxidases by searching KEGG annotation results for keywords "oxidase" and "cytochrome". This analysis yielded no hit in JZ bin_10 or GMQ bin_10, except for two genes: one annotated as "glycolate oxidase" and the other annotated as "Bacterioferritin (cytochrome b1)", suggesting that these organisms are strict anaerobes. Different types of cytochrome c oxidase were detected in the other four genomes, which confirms the previous findings. Complex I of the respiratory chain was observed in all genome bins because they show close relationship to group 4 membrane-bound NiFe

hydrogenases and some subunits of multiple resistance to pH (Mrp) Na⁺/H⁺ antiporters. To address this, we extracted all the Complex I-related genes and their neighbors, and BLASTed them against the NCBI-nr database to retrieve the detailed annotations and conserved domains. Nearly complete complex I was identified in these two anaerobes except *nuoK* and *nuoN*. However, the identified *mrpC* and *mrpD* show homology to these two subunits (Mathiesen et al., 2002), which means they may bear a different function. We further conducted an investigation on *nuoD* to check if they possessed the CXXC binding motifs. The answer was affirmative, and suggested that the *nuoD* homologues in these two bins should be classified as group 4 NiFe hydrogenases (Marreiros et al, 2013), which could reduce protons to hydrogen (Detailed description please see Revised Manuscript Lines 146-154). Overall, GMQ bin_10 and JZ bin_10 are quite different from the others, hence, we conducted a comparative analysis between all sequenced Aigarchaeota genomes at Lines 213-222 in Revised Manuscript.

Comment 2: Figure 2 as presented is highly misleading, because the presence of a gene in 5 out of 6 or the presence of a gene in a single organism have very different implications in the interpretation of the overall Aigarchaeota metabolism. If some cases are well explained in the text, others are not so clear.

Response: We agree that the previous version of Figure 2 was difficult to interpret and drawn such that in some cases it was difficult to follow the different genomic bins in the text. To make it clearer, we redrew Figure 2 and used different colors to show which genes occurred in all, ≥ 3 , 1-2 and none of the six aigarchaeal genomes (See Revised Figure 2). Also, the genes annotated in each bin has been fully recorded in Supplementary Table 2.

Comment 3: I also missed a full description of the metabolic reconstruction of each one of the six genomes, with their KEGG/COG etc annotations, involvement in which metabolic pathway as well as protein family membership (present in which of the over 6000 MCL protein families, is this gene a singleton). This would allow a full inference of the metabolism of each of the genomes sequenced as well as a better evaluation of the author's conclusions.

Response: The description of metabolic reconstruction has been added and moved to Supplementary information. We mainly used the IMG annotation results to reconstruct the metabolic pathways of the six genomes. Those annotation results were generated based on the individual genes in each genome, not the protein families produced by the MCL algorithm. All the genes shown in Revised Figure 2 are fully described in Supplementary Table 2, with their names, putative functions and occurrence frequencies among six genomes provided.

Comment 4: I have some reservations regarding the inclusion of genomes less than 90% complete into an ancestral gene content reconstruction since the Count analysis might lead to incorrect inferences of gene gains and losses. In any case, with exception of the cases discussed in the text and general categories, information is given regarding the 1085 genes shared by both phyla, the ones count attributed to being present, gained or lost in the ancestor of aigarchaeota and/or in the ancestor of thaumarchaeotal lineages. A list of these genes would be much appreciated and would give better insights on the metabolic evolution of the lineages.

Response: We understand these reservations, and it is true that the incomplete genomes may lead to bias in reconstructing the gene content of the ancestor. To address this, we conducted the same analysis based only on the genomes with completeness > 90% to test whether they show similar gene gain and

loss patterns (see Figure below). Basically, the lower the completeness of the sequenced genome is, the more gene loss events or incorrect ancestors will be detected by COUNT. However, we would have lost a large number of genomes if the completeness threshold was increased to 90%, which may lead to more confusion in the case of the transitional genomes because only one is >90% complete. If only one genome were taken into consideration, we definitely would underestimate the gain events at the root of transitional genomes and the common ancestor of Thaumarchaeota and Aigarchaeota, such as Node 9, 53, 54 and 55 in Revised Figure 3. Also, Nodes 47 and 48 would totally disappear if we use the 90% criterion. Also, Nodes 12 and 13 would be combined into one, which would lead to the lower resolution of evolutionary histories of these two phyla. Moreover, this would lead to overestimation of gene families for each genome (see the figure below and Revised Figure 3), even though it does not have a great influence on the general pattern. This effect is mainly caused by the absence of links provided by those removed genomes. For example, the rBBHs for gene b in genome B are gene a and c in genome A and C respectively, but gene a and c are less similar. Based on the MCL algorithm, a, b and c would cluster into one group despite the lower similarity between a and c. If we remove genome B due to the relative lower quality, gene a and c would cluster into different groups which lead to the gene families for both increased. Finally, we had to lower the criteria to recruit more transitional genomes in order to better understand the evolutionary history of Thaumarchaeota and Aigarchaeota and main roles of those transitional genomes in connecting these two lineages. To make a better understanding of metabolic evolution of the lineages, we conducted a new table in which all the gained or lost events at key nodes including ancestors of Aigarchaeota, Thaumarchaeota and ancestral gene families of the two groups were listed (Table S6). Also, some interesting findings have been added in the Revised Manuscript (Lines 313-353).

Comment 5: Based on large state-of-the-art phylogenetic analysis, the “hot” origin of thaumarchaeota (and in fact, of Archaea as domain) was already proposed (Rayman et al PNAS 2015). This could strengthen even further the evolutive scenario in here proposed.

Response: Yes, we agree that Archaea, even LUCA, are originated from thermal habitats. Here, we added a reference to the paper as supporting evidence to further strengthen our findings (See Revised Manuscript Lines 321-325).

Comment 6: Last, not being too familiar with HGTector, after reading the original paper and the methods in here provided I have a methodological question. From what I understood, the count and HGTector analyses were performed independently. How congruent are they? How many cases have Count attributed the origin (and possibly subsequent loss) of a protein family in the ancestor of a lineage and HGTector identify the same protein family as independent cases of HGTs?

Response: The programs COUNT and HGTector assess the evolutionary history of gene families using two very different, and complementary strategies. They each have advantages and limitations. Together, they provide a more comprehensive, and less biased view of the evolution of Aigarchaeota and Thaumarchaeota.

COUNT takes as input a phylogenetic tree of the host genomes (the species tree), and a table of numbers of copies per gene family per genome, and attempts to reconstruct the ancestral states of gene contents using the parsimony approach. From this result, we learn the putative gain and loss events occurred in certain lineages in the history (Figure 3). However, for the gene gain events, COUNT cannot distinguish among *de novo* origination, duplication or horizontal gene transfer (HGT), nor can it relate them to taxa that are not part of the phylogenetic tree.

HGTector, on the other hand, does not reconstruct ancestral states using an explicit phylogeny. Instead it analyzes the BLAST hit tables of all genes against the reference database (NCBI nr), and attempts to identify “atypical” distribution patterns of matches which are likely caused by HGTs. Further, it provides the extra insight of the sources (donors) of the transferred genes. This is particularly useful in surveying the genetic connections from various taxonomic groups to Aigarchaeota and Thaumarchaeota (Figure 4).

For example (as shown in the schematic diagram below), there is one lineage composed of 12 genomes including A-L with one gene family detected in red individuals. Obviously, at node N1, a gain event occurred, letting the descendants of this lineage acquire this function. However, a subsequent gene loss event occurred at N2, resulting in the deficiency of this function in the sub-lineage. Another loss event occurred at N4, leading to the loss of function for I and J. Gene gain event at N3 make A and B regain this capability. So, there are in total four events, including two gain and two loss events, through the whole lineage for this gene family. However, it is impossible to tell whether they are acquired through HGT or duplication. Even though E, F and G have this gene family, they may inherit it from their parents for E and G, but horizontally transfer it from other species for F. To test whether they acquired the genes through HGT, those individual genes should put into the context of all sequenced homolog genes. Overall, the gene gain and loss events detected by COUNT could explain the past about how one gene family evolved historically. However, HGTector can help understand where they came from by comparing them to extant genes in current existed genomes. Therefore, we thought the outputs of COUNT and HGTector are not directly comparable.

One gene family was detected (●) or undetected (●) in specific genomes

Minor:

Comment 7: Although DRTY7 genome appears in several figures (e.g Figure S2, Figure X), and is marked as a genome from this study, no further information (or I missed it) is provided regarding the quality of this draft.

Response: Changed as suggested (Detailed genomic information of DRTY7 was added in Table 1).

Comment 8: Recently, the ubiquitin system of *C. subterraneum* (Aigarchaeota) was functional characterized. This might be a good reference when addressing the presence of the ubiquitin system within these organisms. Hennell James et al, Nat. Comm, 2017

Response: This is certainly a very informative paper, and we thank the reviewer for drawing it to our attention. In the Revised Manuscript, the potential functions of the ubiquitin system are now described by referring to this paper (See Lines 241-249).

Reviewer #2 (Remarks to the Author):

The authors obtained a total of 6 metagenomic bins belonging to Aigarchaeota from two hot springs in China and discussed their physiology especially for carbon and energy metabolisms. In addition, evolutionary relationship between Aigarchaeota and Thaumarchaeota was discussed. The genomic information of Aigarchaeota, a member of TACK superphylum, has been limited despite its importance in exploring the origin of eukaryotic cellular system and evolution of the domain Archaea. The findings from the 6 bins will contribute to understand the genomic diversity and evolution of Aigarchaeota. However, there are issues through the manuscript described below.

Comment 1: L46: Why did the authors define “energy-deficient”. No data presented about this issue in this manuscript. In energy deficient environment, microbial mat formation is not expected while Aigarchaeota has been detected in microbial mat formation.

Response: We are sorry about the unclear description of this sentence. Here, we meant to describe the low organic carbon content of the hot spring habitats. We have now added data on TOC and DOC to support this statement. Most circumneutral pH and alkaline geothermal environments have low organic

carbon concentrations (Badhai et al., 2015; Siering et al., 2006, 2013). The TOC for Firehole Pool in Yellowstone National Park, for instance, was as low as 2 µg/L. Also, Boiling Spring Lake (BSL) in Lassen Volcanic National Park was defined as oligotrophic ecosystems (Siering et al., 2006, 2013). Some studies found relatively high TOC, which can be attributed to nearby foliage or to the dissolution of organic-rich sedimentary rock and suspension of clay-TOC complexes by acidic spring water. For instance, the Deulajhari hot springs with relatively lower temperature (43~69°C) possessed TOC as high as 38 mg/L (Singh et al., 2016), but two other springs in the area are low, with TOC equal to 3.6 and 9.7 mg/L respectively. It seems that some representatives of Aigarchaeota are oligotrophic and may not respond to high concentrations of carbon sources (Beam et al., 2016). In Beam's work, **low** DOC (~40 µM) was detected in their hot spring filamentous 'streamer' communities. Also, *Candidatus Caldiarchaeum subterraneum* was first found in a geothermal water stream from a subsurface gold mine in Japan. This habitat is a typical oligotrophic habitat. As reported, the accretional basement in the oceanic subduction zone might play an important role in providing ammonium and CO₂ as energy and carbon sources for the members in the community (Hirayama et al., 2005). In our case, organic carbon concentrations are low (TOC: 6.8 and 0.33 mg/g; DOC: 22 and 39 mg/L). To avoid ambiguity, this sentence was revised in the Revised Manuscript (Line 51). Geochemical data was added if available (Revised Manuscript, Supplementary Table 1).

Comment 2: L66: Nunoura et al. 2010 is likely the first report detecting aigarchaeal SSU rRNA gene sequence from a deep-sea hydrothermal environment.

Response: We thank the reviewer for this suggestion, and now cite that paper in the introduction.

Comment 3: L83-: I could not identify the relationship between the metagenomic bins and metagenomic libraries through the text, tables and figures through the manuscript including supplementary materials whereas a total of 5 samples were sequenced (L327). Did the authors co-assemble multiple shotgun libraries? Bowers et al. 2017 should be referred for evaluating metagenomic bins.

Response: We apologize for the unclear description. We did not co-assemble multiple shotgun libraries. Rather, we conducted metagenomic analysis on five hot spring samples. Only two of them have Aigarchaeota genomes. So, in this study, "five" samples should be changed to "two" (Revised Manuscript, Lines 412 and 418). The paper written by Bowers et al. (2017) was used to evaluate the quality of the metagenomic bins and is cited in the Revised MS (Line 100).

Comment 4: L99-: Takami et al. 2015 should be referred in this paragraph.

Response: Done as suggested (Revised Manuscript, Line 146).

Comment 5: L100-: Cellulase is very diverse and its characterization is very complicated. The information described here is not sufficient to characterize their properties especially the one manual curation was required.

Response: That's right. Cellulases are very diverse and complicated. Here, we add one figure and one table to talk about the carbohydrate-active genes based on the annotation with CAZy database (Supplementary Table 3). Further, the conserved domains of the endoglucanase were predicted and compared to other putative or verified cellulases (see Supplementary Fig. 5). Still, we cannot

confidently infer they are cellulases. Hence, we conducted a prediction of their protein structure using SWISS-MODEL (Biasini et al., 2014), and found that this protein will likely form a compact dodecamer in solution as shown below. Two Trp residues on the surface were identified on the predicted dodecamer, which may serve as binding sites to cellulose and polysaccharides (Sakon et al., 1997). We are still not convinced this is conclusive, so we did not put this in the Revised Manuscript.

Figure 1. The overall structure of endoglucanase in JZ bin_40. (a) The dodecamer formed in the crystal by the assembled symmetry-related monomers. (b) Two identified Trp residues. Both of them are located on the surface of the dodecamer. The figure is generated by ICh3D (NCBI Resource Coordinators, 2017S).

Comment 6: L106-: New line is required for the discussion of carbon fixation pathways.

Response: Revised as suggested (see Line 134 in Revised MS).

Comment 7: L106: The proposal of a new rTCA cycle described in Goltsman et al. 2009 is interesting, but both enzymatic activity measurements and metabolomics are absent. Thus, the authors should repress the interpretation of the potential rTCA cycle in Aigarchaeota.

Response: Previously, our manuscript lacked a compelling argument that this potential pathway is workable. However, in a recent study, *Thermosulfidibacter takaii* ABI70S6 was shown to fix carbon dioxide under chemolithoautotrophic conditions through the rTCA cycle with the highly efficient and reversible citrate synthase (Nunoura et al., 2018). Therefore, we referred this paper as evidence to strengthen our interpretation (see Revised Manuscript, Lines 137-141).

Comment 8: L114-115: What does *nuo* operon mean? Appropriate reference should be provided after aerobic.

Response: *nuo* is short for NADH ubiquinone oxidoreductase. This gene encodes the subunits of complex I of the respiratory chain. We are sorry that we made a mistake in clarifying if they are aerobes. The *nuo* operon is not an indicator of aerobic respiration, which has been pointed out by reviewer #1. Instead, heme-copper terminal oxidase could be used to judge if microbes could use oxygen as the electron acceptor. We modified related description in Revised Manuscript (Line 146-154).

Comment 9: L127: Sulfur and hydrogen are important energy source even in aerobic environments.

Response: Sulfur and hydrogen are indeed quite important to thermal habitats under both anaerobic and aerobic conditions. We rewrote this sentence in the Revised Manuscript (Line 167).

Comment 10: L134-: “this study” should be deleted. Only one example of the potential HGT is discussed here. The manuscript discussed only aigarchaeal metagenomic bins and no data was presented about HGT in the hot spring environment.

Response: Revised as suggested (Line 174 in the Revised Manuscript).

Comment 11: L150: Homologues of heterodisulfide reductase are often identified in genomes of non-methanogenic anaerobes although their functions are unknown in most of the cases. It is very curious to mention “By consuming the CoM-SS-CoB generated by methanogens”. Did the authors have any evidence about it? It is not allowed based on the data set presented in this manuscript.

Response: After careful reconsideration about this, we now realize that there is insufficient evidence to support this inference. According to our literature search, the function of the *hdr* complex has been reported in several previous studies (Mander et al., 2004; Ramos et al., 2015). In combination with previous findings, we propose that the *hdr* complex may play an essential role in transferring electrons for this sulfite-reducing microorganism. By coupling with *mvh* complex, H₂ was oxidized and protons and electrons were released. Then electrons were transferred to DsrC and the generated reduced DsrC may act as electron donors to promote the reduction of sulfite. Detailed as described in Revised Manuscript (Lines 186-192).

Comment 12: L173-: Hennel et al. 2017 should be referred.

Response: Maybe you mean James et al., 2017? If yes, added as suggested (Revised Manuscript, Line 249).

Comment 13: L173-: After the paragraph, other aigarchaeal genomes obtained in other study should be combined because the three paragraphs discussed general genomic features of this phylum.

Response: Done as suggested. We made a comparison among current available Aigarchaeota genomes (detailed as described in Revised Manuscript Line 210-234).

Comment 14: L177: Hershko and Ciechanover 1998 is the reference for eukaryotic ubiquitin system, and appropriate reference of aigarchael ubiquitin system should be presented. In addition, I think Aigarchaeota harbor E3-like protein but not E3 protein.

Response: Revised as suggested. I think James et al., 2017 should be suitable here (Line 249 in Revised Manuscript)

Comment 15: L185: Delete “other”.

Response: Revised as suggested (Line 257 in Revised Manuscript)

Comment 16: L197-: The genome information of *Nitrosocaldus* (Abby et al. 2018) should be added in the revision.

Response: Combined with the opinions from reviewers #2 and #3, they both pointed out the importance of *Nitrosocaldus* in the evolution of Thaumarchaeota and most of them were identified to possess the capability of ammonia oxidation. Therefore, two genomes (*Candidatus Nitrosocaldus islandicus* isolate 3F and *Candidatus Nitrosocaldus cavascurensis* strain SCU2) and one species with a 16S rRNA gene but no genomic information (*Nitrosocaldus yellowstonii* strain HL72) belonged to this genus were added in the Revised Manuscript (Figure 1c, Supplementary Figure 2 and Supplementary Table 4).

Comment 17: L205-: Did the authors test concatenated RNA polymerase tree?

Response: As suggested, we reconstructed the RNA polymerase tree based on 10 concatenated RNA polymerase genes including *rpoA1* (K03041), *rpoB* (K13798), *rpoD* (K03047), *rpoF* (K03051), *rpoH* (K03053), *rpoN* (K03058), *rpoK* (K03055), *rpoL* (K03056), *rpoE1* (K03049), and *rpoP* (K03059). Other RNA polymerase genes were unusable for this analysis because they were absent from more than half the genomes. The same criteria were employed to reconstruct the RNA polymerase tree and the r-protein tree. Compared to the ribosomal protein phylogeny, we observed little discrepancy between them. The two draft genomes Aigarchaeota archaeon JGI MDM2 JNZ-1-N15 and Aigarchaeota archaeon JGI MDM2 JNZ-1-K18, which were isolated through the single cell technology, showed closer phylogenetic distance to Thaumarchaeota in the RNA polymerase phylogeny, with high bootstrap confidence (>70%). However, they were previously classified as Aigarchaeota based on the 16S and concatenated ribosomal proteins, but the bootstrap confidence for the root of Aigarchaeota was not high. In addition, the authors who isolated them also named them as Aigarchaeota. (The sequenced genomes were uploaded to the IMG database, but no published papers are available.) The discrepancies further implicate the close relationship between these two phyla. Related discussions including discrepancies were described in the Revised Manuscript (Lines 283-291).

Comment 18: L222-: Information of Nitrosocaldales represented by Nitrosocaldus and of pSL12 lineage should be provided. pSL12 lineage (also called 1.1c etc. e.g. Weber et al.) is also important in the physiological evolution of Thaumarchaeota because it probably lack capability of ammonia oxidation.

Response: As suggested, genomes in HWCIII group including *Candidatus Nitrosocaldus islandicus* isolate 3F (CP024014.1) and *Candidatus Nitrosocaldus cavascurensis* strain SCU2 (LT981265.1) have been added in the Revised Manuscript. Also, other members with the existence of 16S rRNA genes but no genomic information like pSL12 group, HWCIII (*Candidatus Nitrosocaldus yellowstonii* strain HL72), group1.3, group 1.1c and SAGMGC-1 group, were also included (see Revised Figure 1, Supplementary Figure 2 and Supplementary Table 4). pSL12 has been found not only in hot spring ecosystems (Barns et al., 1996), but also open ocean habitats (Mincer et al., 2007). However, Pester et al. (2011) treated them as ALOHA lineage which may possess the ammonia oxidation ability. Currently no genomics information has been reported for the pSL12 lineage in hot springs, so there are not data to predict if they are ammonia oxidizers or not. It needs to be mentioned that different papers used different classification criteria for the thaumarchaeal groups. Group 1.1c and pSL12 were treated as different groups in the review conducted by Pester et al. (2011). We find that Nitrosocaldus play a key role on the evolutionary history of Thaumarchaeota (Node 10 in Revised Figure 3. Detailed discussion was described in Revised Manuscript Lines 335-353).

Comment 19: L230-: The calculation of the divergence age is interesting but is not important in this manuscript because discussion about the paleoenvironmental information was absent in this manuscript.

Response: As the same view by Reviewer #3, they both pointed out it is unnecessary to talk about divergence time. Besides, the calculation of the divergence is problematic due to the lack of cross-validation and fossil calibration. So, finally we decided to remove this analysis from the Revised Manuscript.

Comment 20: L259-: The number of aigarchaeal genomes is not sufficient to discuss a genome streamlining pattern.

Response: Done as suggested, we revised that paragraph and the discussion about genome streamlining was deleted (See Line 355-357 in Revised Manuscript). But we retained the description about the SNP pattern because it's an interesting finding that GMQ bin_10 and JZ bin_10 have tremendous differences in SNPs even though they show high genomic similarity.

Comment 21: Fig. S2: This phylogenetic tree did not refer previous reports in naming of each group. For example, in the case of Marin Group I (also called 1.1a), grouping has been proposed in Massana et al. (2000), Takai et al. (2004) and Lauer et al. (2016). The information about Nitrosocaldus and pSL12 group was also absent as described above. Stiegmeier et al (2014) in The prokaryotes- Other major lineages of Bacteria and the Archaea is also helpful.

Response: For the original Manuscript, we classified them into different groups/subgroups based on the sequence similarity of the 16S rRNA gene. As suggested, here we changed the group names according to Pester et al. (2011, Current Opinion in Microbiology). We also added several 16S rRNA gene sequences to Figure S2, so now most thaumarchaeal families are included in the tree (detailed as described in Comment 18). After we reconstructed the 16S-based phylogenetic tree, we were surprised to realize that DRTY7 bin_36 in this study might be the first genome of the deep-branching pSL12 group.

Reviewer #3 (Remarks to the Author):

Overall, I find this to be a very interesting and well conducted study that will be a valuable contribution to our understanding of the environmental niche of a poorly understood phylum of archaea.

Comment 1: I have, however, reservations about how some of the phylogenetic analyses were conducted or interpreted. That part of the MS needs to be revised before the MS is acceptable for publication. In particular, I have problems with the concept of deriving divergence times from phylogenetic trees of genes/organisms for which no cross-validation exists. Such analyses rest on the idea that it is possible to extrapolate validated "hard" fossils (that only date back a few hundred million years) into the Hadean and Archean past (2.5-4 billion years ago). There is no scientific support for this concept. On the contrary, past studies have shown that different genes' and organisms' "molecular clocks" "tick" at different intervals. This observation, together with the fact that the concept that today's organisms/genes/habitats are representative of genes/lifeforms/environments that existed billions of years ago is an untestable hypothesis, makes the extrapolation from today's genes/organisms into the past not only speculative but unscientific. I strongly urge the authors to remove any of this data from their ms, because it weakens an otherwise great study.

Response: We sincerely thank the reviewer for the positive assessment of our study and truly appreciate such a detailed introduction about the estimation of divergence times. Combined with comments from Reviewer #2, we decided to remove this part from the Revised Manuscript.

Comment 2: In addition, I ask the authors to please provide more environmental metadata that would help us to better interpret some of the findings in the genomes, especially the ones that relate to

elemental cycling and interactions with the environment and other organisms. The bare minimum would be a description of the temperature, pH, TOC, DOC, DIC, DON, and most important cat- and anions (trace metals would be good but are not considered a must) in the two springs. Was there grass growing around the springs that could provide a source of cellulose/lignin? Ideally, this data would then be put into context to the environments from which the other 8 published aigarchaeotal genomes have been obtained.

Response: Thanks for the valuable suggestion. We have measured a series of physical and chemical parameters of the two sampling sites as listed in Supplementary Table 1, and provided detailed sample information in Supplementary Methods (Lines 3-8). GMQ has some nearby vegetation, and some leaves have been seen in the source pool and outflow channel. JZ is a geothermal well that sometimes accumulates some leaves as well. In addition, bacteria such as Cyanobacteria in the springs might also produce cellulose and other exopolysaccharides. We agree with the general idea to better leverage the physicochemical data to understand these organisms, but this might better be done within the context of a larger survey for Aigarchaeota, which we are currently conducting using specific PCR primer sets. We hope to combine information from the PCR surveys with the genomic information here to better understand these different Aigarchaeota in an environmental context.

Detailed comments

Comment 3: I suggest to shorten the title to read "Genomic inference of the metabolism and evolution of the archaeal phylum Aigarchaeota"

Response: Revised as suggested.

Comment 4: L223-226 and 237-239, this is not exactly true. Hatzenpichler2008 identified the first thermophilic ammonia oxidizer, a discovery that was soon followed up by the report of *Nitrosocaldus yellowstonii*, which is adapted to much higher temperatures. Hatzenpichler postulated "that archaeal ammonia oxidation evolved under thermophilic conditions with the mesophilic lifestyles exemplified by soil or marine AOA likely representing independent, secondary adaptations to lower temperatures." This idea was later supported by the idea of a thermophilic origin of Thaumarchaeota in general, see for example:

Brochier-Armanet C, Gribaldo S, Forterre P. 2012. Spotlight on the Thaumarchaeota. *ISME J.*;
de la Torre JR, Walker CB, Ingalls AE, Könneke M, Stahl DA. 2008. Cultivation of a thermophilic ammonia oxidizing archaeon synthesizing crenarchaeol. *Environ. Microbiol.*;
Groussin M, Gouy M. 2011. Adaptation to environmental temperature is a major determinant of molecular evolutionary rates in archaea. *Mol. Biol. Evol.*
Nunoura T, et al. 2011. Insights into the evolution of Archaea and eukaryotic protein modifier systems revealed by the genome of a novel archaeal group. *Nucleic Acids Res.*

Response: After genomes from genus *Nitrosocaldus* were taken into consideration, we totally agree with this argument. From the revised phylogenetic tree of Thaumarchaeota and Aigarchaeota (Revised Figure 3), we found that AOAs evolved from thermal habitats and *Nitrosocaldus* appears to be the ancestor of mesophilic AOAs. *Nitrosocaldus* plays a critical role in helping us understand the

evolutionary history of AOAs. We modified this in the Revised Manuscript (Line 321-343) and cited papers listed above. Moreover, the urea oxidation and cobalamin biosynthesis pathways were also evolved from thermal habitats, which may improve their competitive advantages when they dispersed to marine and soil habitats (see Revised Manuscript Lines 335-353).

Comment 5: L233-234 and 239-241, timing of evolutionary events cannot be inferred from phylogenetic trees alone but needs external validation; the only study that I am personally aware of that ever achieved such validation was a study on bacterial insect-symbionts. In that case the timing could be cross-validated with insect fossils.

While some studies - including the one by David and Alm - claim to reliably date the timing of major evolutionary events, these are in fact untested (often even untestable) hypotheses. They all rest on the idea that it is possible to extrapolate validated "hard" fossils (that only date back a few hundred million years) into the Hadean and Archean past (3-4 billion years ago). Neither is there support for this idea, nor is there support for the concept that "molecular clocks" indeed "tick" in a steady interval and that they do not differ between different organisms or genes living in different habitats.

The authors also do not discuss why they choose the divergence time of *Sulfolobus*/*Aeropyrum* and not any other group for calibration. In the end, it doesn't really matter because all these divergence times are mere speculation. The study by David and Alm is flawed, because the data features not only many misalignments but also rests on the (flawed) comparison of paralogues (rather than homologous) sequences/protein families.

Response: We thank the reviewer for this suggestion. After careful reconsideration, we decided to delete this part from the manuscript, following the suggestion by Reviewer #2 and #3.

Comment 6: Fig3, I suggest to either completely remove the figure from the MS or keep the underlying tree but erase the divergence dates from the tree; besides the fact that I have severe doubts about putting time stamps onto phylogenetic trees, I am surprised that the tree is so well resolved and does not show any multifurcations. Please indicate the Bayesian values for every node, and discuss why you chose not to collapse nodes with low Bayesian support values

Response: As described above, we decided to remove the divergence time estimates. For the Bayesian tree, it is well resolved with all posterior probabilities > 0.9 . Previously, we used phyloBayes (Lartillot et al., 2009) to construct the Bayesian tree, because it is widely used for divergence time estimation. In this version, since the divergence time estimation is removed, we used the more generally used MrBayes (we added two more genomes belonging to genus *Nitrosocaldus*, so it was necessary to reconstruct the tree). Results show that the re-generated Bayesian tree closely resembles the previous one, with all the Bayesian support values > 0.9 (see Supplementary Figure 18 with posterior probabilities listed on each node).

Comment 7: FigS2 and S6, bootstrap support $>50\%$ is the bare minimum; please indicate only nodes $>70\%$ and $>90\%$ support; all other nodes should be collapsed because they must be considered unreliable.

Response: Thanks for the suggestion. We modified nearly all the phylogenetic trees including Revised Figure 1, Figs S2, S7, S15. Different colors were used to show the bootstrap confidence of each node with greens for $> 70\%$ and reds for $> 90\%$ support. However, for the presentation purpose, we didn't collapse nodes for the unreliable nodes for Revised Figs S6 and S10 due to the large quantity of leaves

in these two trees.

Comment 8: It is not documented how phylogenetic analyses of DsrAB, cox, and nif genes/proteins were performed and how bootstrapping support values were derived.

Response: Done as suggested, details of the tree reconstruction approach were described in Supplementary Methods (Lines 39-71).

Comment 9: In terms of the two genes predicted to be involved in cellulose degradation, please discuss if any other genes suggest the involvement of Aigarchaeota in cellulose degradation; how would these enzymes be transported outside of the cell and how would mono/di saccharides reach the inside of the cell? are any other archaea known to be involved in cellulose degradation? Only bacteria come to mind, but I do not appreciate the latest literature.

Response: It's true that few archaeal species are known to degrade cellulosic biomass, especially for those hyperthermophilic Archaea. However, we found several studies reported that several microbes could grow on cellulosic substrates. For example, a hyperthermophilic archaeon, *Desulfurococcus fermentans*, demonstrated growth on crystalline cellulose at an optimum temperature of 81 °C (Perevalova et al., 2005). Also, another study showed that one community composed by several hyperthermophiles could deconstruct lignocellulosic biomass at 90 °C (Graham et al., 2011). In that community, for the first time, they found a multidomain cellulase in the dominant strain, which are ubiquitous among cellulolytic microbes but nearly no report on hyperthermophilic archaea (Graham et al., 2011). By comparing to the CAZy database, we found that glycosyl hydrolases (GHs) are enriched in JZ bin_40 including 29 carbohydrate-degradation genes distributed in 15 GHs. Also, 26 genes related to oligosaccharide transporters have been identified in this bin. These findings, to some extent, support our opinion that some microbes could utilize those complex carbon sources (see Revised Manuscript Lines 119-133).

References

- Badhai, J., et al. Taxonomic and functional characteristics of microbial communities and their correlation with physicochemical properties of four geothermal springs in Odisha, India. *Front Microbiol.* **6**, 1166 (2015).
- Beam, J. P. et al. Ecophysiology of an uncultivated lineage of Aigarchaeota from an oxic, hot spring filamentous 'streamer' community. *ISME J.* **10**, 210-224 (2016).
- Graham, J. E., et al. Identification and characterization of a multidomain hyperthermophilic cellulase from an archaeal enrichment. *Nat. Commun.* **2**, 375 (2011).
- Hirayama, H., et al. Bacterial community shift along a subsurface geothermal water stream in a Japanese gold mine. *Extremophiles* **9**, 169-184 (2005).
- Lartillot, N., Lepage, T. & Blanquart, S. PhyloBayes 3: a Bayesian software package for phylogenetic reconstruction and molecular dating. *Bioinformatics* **25**, 2286-2288 (2009).
- Mander, G. J., Pierik, A. J., Huber, H., & Hedderich, R. Two distinct heterodisulfide reductase-like enzymes in the sulfate-reducing archaeon *Archaeoglobus profundus*. *FEBS J.* **271**, 1106-1116 (2004).
- Perevalova, A. A., et al. *Desulfurococcus fermentans* sp. nov., a novel hyperthermophilic archaeon from a Kamchatka hot spring, and emended description of the genus *Desulfurococcus*. *Int. J. Syst. Evol. Microbiol.* **55**, 995-999 (2005).
- Ramos, A. R., et al. The FlxABCD - HdrABC proteins correspond to a novel NADH dehydrogenase / heterodisulfide reductase widespread in anaerobic bacteria and involved in ethanol metabolism in *Desulfovibrio vulgaris* Hildenborough. *Environ Microbiol* **17**, 2288-2305 (2015).
- Sakon, J., Irwin, D., Wilson, D. B. & Karplus, P. A. *Nat. Struct. Mol. Biol.* **4**, 810-818 (1997).
- Siering, P. L., et al. Geochemical and biological diversity of acidic, hot springs in Lassen Volcanic

- National Park. *Geomicrobiol. J.* **23**, 129-141 (2006).
- Siering, PL., et al. Microbial biogeochemistry of Boiling Springs Lake: a physically dynamic, oligotrophic, low-pH geothermal ecosystem. *Geobiology* 11, 356-376 (2013).
- Singh, A., & Subudhi, E. Profiling of microbial community of Odisha hot spring based on metagenomic sequencing. *Genom. Data* 7, 187-188 (2016).

REVIEWERS' COMMENTS:

Reviewer #1 (Remarks to the Author):

In the revised manuscript, the authors took into consideration all of the reviewers concerns and have redone most of the comparative analysis.

This is an improved version of the manuscript that covers many relevant aspects regarding Aigarchaeota (and Thaumarchaeota evolution).

Minor:

- Please replace throughout the text sulfur oxidation for sulfide oxidation. the sqr oxidizes sulfide to elemental sulfur/polysulfides.

Reviewer #2 (Remarks to the Author):

Major comments

The contents have been significantly updated comparing to the previous version of the manuscript. However, I found points that should be revised prior to the publication.

L40: "Microbes of the phylum Aigarchaeota are abundant in hot spring sediments". This is not general role in Aigarchaeota, and the sentence should be reconsidered.

L45: "strains" is not appropriate. "genome bin" or other appropriate word should be used instead of "strains".

L51-: In the phylogenetic tree of Thaumarchaeota, Nitrososphaerales inhabiting mostly soil and sediments is placed between Nitrosocaldales and Nitrosopumilales. Mesophilic environments include soil, sediments and water, and thus the availability of ammonia is diverse among these mesophilic environments. Thus, the sentences are not appropriate.

L66: terrestrial and subsurface geothermal systems and marine hydrothermal environments.

L80: Results and discussions

L90: contamination of other genome fragments

Fig. 2: rTCA cycle is not appropriate. If glycolysis occurs, rTCA cycle with citrate synthase probably does not occur. The author should use TCA cycle, reversible TCA cycle or roTCA cycle.

If Aigarchaeota operates the rTCA cycle with citrate synthase under aerobic condition, it is necessary that succinate dehydrogenase must not be involved in the aerobic respiratory chain. However, the figure indicates that succinate dehydrogenase is a part of aerobic

respiratory chain. This point should also be discussed in main text.

L118: What does "the best hit for one of the genes"? Did it define by e-value, identity or other factors?

L124: Both ATP citrate lyase and citryl-CoA synthase/citryl-CoA lyase were absent,

L124-: I think it is better to place results and discussions for the potential of chemolithoautotrophy before discussing the potential of carbon fixation in Aigarchaeota.

L127-: roTCA cycle in *Desulfurella* in Mall et al. (2018) should also be cited.

L130-: The sentence from "This protein might represent" should be deleted.

L211-: If the authors discuss about the "oligotrophic habitats" of Aigarchaeota, the discussion should be focused on heterotrophic metabolisms because sufficient amount of reduced inorganic compounds for chemolithotrophy is usually supplied in the geothermal environments where the genomes have been retrieved.

L227: Several eukaryotic signature proteins were identified in Aigarchaeota genomes as described before (Refs).

L260-303: The discussion in this paragraph is quite complex. Why did not the authors discuss about the phylogeny based on the 16S and 23S rRNA gene sequence from the same data set here? How about the distribution patten of core genes?

L261: Hot or high temperature is better than thermophilic

L286: mesic? It should be "cold or ambient temperature"

L304-: Discussions about the evolution of Thaumarchaeota after divergence of Nitrosocaldales and pLS12 should be deleted. The data presented here is not sufficient about the issue and it is not the major focus of this manuscript.

Reviewer #3 (Remarks to the Author):

with the improvements done by the authors this is now a very strong ms that I am sure will be widely read by the research community

please change the following:

L298-303, This evolutionary scenario has been proposed before using crenarchaeol and other lipid markers as molecular evidence; please cite the relevant literature

Fig1, please add T and pH information on the hot springs into panelA

"Thaumarchaeota archaeon N4" has been given a Candidatus name, *Cand. Nitrosotenuis uzonensis* (Lebedeva2013); please change ms and figs accordingly

FigS2, delete "Nitrocosmicus" f(under group Ib) from the figure; that genus does not exist

Fig 3, it is unclear what the red circles indicate; you state in your rebuttal letter that all nodes were >0.9 supported, so it's not Bayesian values. What are these?

I also first didn't understand what the green is supposed to indicate; it would be easier to draw lines from the insert to the dashed box so that's it's clear that this is a zoomed-in view of this thaumarchaeotal branch

There is a lot of red-green color blind people out there; I suggest to change the colors of the support nodes in your trees (red-blue is typically considered a good combination)

Also, please add the information on what each colored node means into either the figure legend or the figure itself (at least for figs3 and s7 this is not stated)

Responses to the reviewer's comments

Reviewer #1 (Remarks to the Author):

In the revised manuscript, the authors took into consideration all of the reviewers' concerns and have redone most of the comparative analysis. This is an improved version of the manuscript that covers many relevant aspects regarding Aigarchaeota (and Thaumarchaeota evolution).

Minor:

Comment 1: Please replace throughout the text sulfur oxidation for sulfide oxidation. the *sqr* oxidizes sulfide to elemental sulfur/polysulfides.

Response: Done as suggested.

Reviewer #2 (Remarks to the Author):

Major comments

The contents have been significantly updated comparing to the previous version of the manuscript. However, I found points that should be revised prior to the publication.

Comment 1: L40: "Microbes of the phylum Aigarchaeota are abundant in hot spring sediments". This is not general role in Aigarchaeota, and the sentence should be reconsidered.

Response: Done as suggested. This sentence was revised as "Microbes of the phylum Aigarchaeota are widely distributed in geothermal environments" (see Revised Manuscript Line 24).

Comment 2: L45: "strains" is not appropriate. "genome bin" or other appropriate word should be used instead of "strains".

Response: Revised as suggested by changing it to "genome bins" (see Revised Manuscript Line 25).

Comment 3: L51-: In the phylogenetic tree of Thaumarchaeota, Nitrososphaerales inhabiting mostly soil and sediments is placed between Nitrosocadales and Nitrosopumilales. Mesophilic environments include soil, sediments and water, and thus the availability of ammonia is diverse among these mesophilic environments. Thus, the sentences are not appropriate.

Response: Thanks for the suggestion. Due to the word limitation, we removed this sentence because we considered this as a known finding.

Comment 4: L66: terrestrial and subsurface geothermal systems and marine hydrothermal environments.

Response: Revised as suggested (see Revised Manuscript Line 39).

Comment 5: L80: Results and discussions.

Response: Revised as suggested (see Revised Manuscript Line 59).

Comment 6: L90: contamination of other genome fragments

Response: Revised as suggested (see Revised Manuscript Line 69).

Comment 7: Fig. 2: rTCA cycle is not appropriate. If glycolysis occurs, rTCA cycle with citrate synthase probably does not occur. The author should use TCA cycle, reversible TCA cycle or roTCA cycle.

If Aigarchaeota operates the rTCA cycle with citrate synthase under aerobic condition, it is necessary that succinate dehydrogenase must not be involved in the aerobic respiratory chain. However, the figure indicates that succinate dehydrogenase is a part of aerobic respiratory chain. This point should also be discussed in main text.

Response: Exactly right as described by Reviewer #5. In our case, the potential carbon fixation pathway should be roTCA rather than rTCA cycle due to the absence of citrate lyase and citryl-CoA synthesis and lyase. This was revised in the Revised Manuscript. However, we are quite confused that why Reviewer questioned that "If Aigarchaeota operates the rTCA cycle with citrate synthase under aerobic condition, it is necessary that succinate dehydrogenase must not be involved in the aerobic respiratory chain". Generally, rTCA or the newly reported roTCA cycle are mainly founded in anaerobic organisms. However, exceptions

also exist in other studies that those pathways are functional in some microaerobic or aerobic microbes such as *Hydrogenobacter thermophilus* (Shiba et al., 1985), "*Candidatus Nitrospira defluvii*" (Lücker et al., 2010) and some *Leptospirillum* genomes (Goltsman et al., 2009). Also, in the study conducted by Lücker et al. (2010), they also found complex I, II, III, IV and V pathways in "*Candidatus Nitrospira defluvii*". Related discussion was added in the Revised Manuscript (Lines 104-109 and 183-187). Thus, to respond to the reviewer better we would need a more detailed criticism.

Comment 8: L118: What does “the best hit for one of the genes”? Did it define by e-value, identity or other factors?

Response: The best hit for it is GBC70340.1 with 99% in query coverage and 86% in amino-acid identity (see Revised Manuscript Line 97). It's noteworthy that the second hit is CDG15325.1, showing 98% coverage and 49% identity. This gene was described as beta-glucosidase which shows activity at hot high temperature (Schroeder et al., 2014).

Comment 9: L124: Both ATP citrate lyase and citryl-CoA synthase/citryl-CoA lyase were absent.

Response: Right, this sentence was revised as suggested (see Revised Manuscript Line 103).

Comment 10: L124-: I think it is better to place results and discussions for the potential of chemolithoautotrophy before discussing the potential of carbon fixation in Aigarchaeota.

Response: Thanks for the suggestion. However, we did not make changes about this because we think the current structure is more clear than the changes as suggested. Generally, the paragraph structure for the manuscript is like this: (1) Carbon cycle: We first describe the glycolysis and TCA cycle, then carbon fixation via the roTCA cycle (it seems reasonable that we discuss the roTCA cycle after the description of TCA cycle) and finally the carbon monoxide oxidation pathway. (2) Sulfur and hydrogen related pathways. Sulfur metabolisms mainly contain sulfide oxidation and sulfite reduction. The sulfite reduction pathway may couple with the H₂ oxidation. Hence, we spent one paragraph to discuss the hydrogenases in Aigarchaeota after the description of sulfur cycle. If we place the carbon fixation behind the discussion of chemolithoautotrophy, the manuscript seems to be messy because we mentioned TCA or roTCA in several above paragraphs.

Comment 11: L127-: roTCA cycle in Desulfurella in Mall et al. (2018) should also be cited.

Response: Done as suggested (see Revised Manuscript Line 106).

Comment 12: L130-: The sentence from “This protein might represent” should be deleted.

Response: Done as suggested.

Comment 13: L211-: If the authors discuss about the “oligotrophic habitats” of Aigarchaeota, the discussion should be focused on heterotrophic metabolisms because sufficient amount of reduced inorganic compounds for chemolithotrophy is usually supplied in the geothermal environments where the genomes have been retrieved.

Response: Thanks for the comment. Actually, we also think the discussion about “oligotrophic habitats” is not useful in the context, so we deleted this part from the Revised Manuscript.

Comment 14: L227: Several eukaryotic signature proteins were identified in Aigarchaeota genomes as described before (Refs).

Response: Done as suggested (see Revised Manuscript Line 196).

Comment 15: L260-303: The discussion in this paragraph is quite complex. Why did not the authors discuss about the phylogeny based on the 16S and 23S rRNA gene sequence from the same data set here? How about the distribution pattern of core genes?

Response: The main point of this paragraph is to reveal phylogenetic relationship of these two phyla. To address this, we conducted the phylogenetic analysis based on both the 16S rRNA gene sequences and concatenated ribosomal proteins. Also, as suggested by Reviewer #2, we reconstructed concatenated RNA polymerase tree to assist explain this question. We did like this because: (1) Phylogenetic analysis based on single genes (e.g. 16S rRNA gene) lead to low bootstrap values of the tree, which may result in an unreliable tree; (2) Some groups only have 16S rRNA gene sequences and no genomes (e.g. ALOHA and

1.3 group). Also, among the collected genomes, 16S rRNA gene sequences might be undetectable. That is why we combine different gene sets to unravel the phylogenetic position of Thaumarchaeota and Aigarchaeota. All evidence suggests that these two phyla are diverse groups. However, based on whole genome comparison, we find an interesting thing that some genomes (the so-called transitional genomes) belong to Thaumarchaeota show closer distance to Aigarchaeota and all those genomes from hot habitats. So we propose the evolutionary scenario of hot origin of these two phyla and lead to the later discussion of their evolutionary relationship. We also think analysis of core genes of different lineages of Aigarchaeota would be interesting and informative, but we think this analysis would be much more interesting when there are more genomes within the Aigarchaeota.

Comment 16: L261: Hot or high temperature is better than thermophilic

Response: Done as suggested (see Revised Manuscript Line 227).

Comment 17: L286: mesic? It should be “cold or ambient temperature”

Response: Done as suggested (see Revised Manuscript Line 251).

Comment 18: L304-: Discussions about the evolution of Thaumarchaeota after divergence of Nitrosocaldales and pLS12 should be deleted. The data presented here is not sufficient about the issue and it is not the major focus of this manuscript.

Response: Done as suggested. In this paragraph, most discussions are described at the key node 10 which is preceded the divergence of Nitrosocaldus and pSL12. This mainly includes genes related to ammonia oxidation, urea metabolism and cobalamin synthesis. Others are deleted in the Revised Manuscript.

Reviewer #3 (Remarks to the Author):

with the improvements done by the authors this is now a very strong ms that I am sure will be widely read by the research community

please change the following:

Comment 1: L298-303, This evolutionary scenario has been proposed before using crenarchaeol and other lipid markers as molecular evidence; please cite the relevant literature.

Response: Done as suggested (see Revised Manuscript Lines 261-263).

Comment 2: Fig1, please add T and pH information on the hot springs into panelA.

Response: Revised as suggested (see Revised Figure 1).

Comment 3: “Thaumarchaeota archaeon N4” has been given a Candidatus name, *Cand. Nitrosotenuis uzonensis* (Lebedeva 2013); please change ms and figs accordingly.

Response: Revised as suggested.

Comment 4: FigS2, delete “Nitrocosmicus” (under group Ib) from the figure; that genus does not exist

Response: After searching the NCBI database, we find that the “*Candidatus Nitrocosmicus oleophilus*” does exist and the taxonomy ID for it is 1353260. Also, Jung et al. published a paper titled “A hydrophobic ammonia-oxidizing archaeon of the Nitrosocosmicus clade isolated from coal tar-contaminated sediment” and proposed the species name as “*Candidatus Nitrocosmicus oleophilus MY3*” (Jung et al., 2016).

Comment 5: Fig 3, it is unclear what the red circles indicate; you state in your rebuttal letter that all nodes were >0.9 supported, so it's not Bayesian values. What are these? I also first didn't understand what the green is supposed to indicate; it would be easier to draw lines from the insert to the dashed box so that it's clear that this is a zoomed-in view of this thaumarchaeotal branch.

Response: Sorry for the unclear description. Actually, the red circles are just node numbers from one to 55 which facilitate us better describe the evolutionary events at the key nodes. For the Bayesian values of this tree, please see Supplementary Figure 18 and you can find that all nodes are > 0.9 supported. For the green arrow, exactly as your understanding that it represents the zoomed-in view of that lineage, as suggested, we revised this figure as the Revised Figure 3.

Comment 6: There is a lot of red-green color-blind people out there; I suggest to change the colors of the support nodes in your trees (red-blue is typically considered a good combination). Also, please add the information on what each colored node means into either the figure legend or the figure itself (at least for figs3 and s7 this is not stated).

Response: Done as suggested. To avoid the red-green color problem, Figs. 1, 2, 4 and Supplementary Figs. 2, 7, 10, 14, 15, 16 were revised. Also, the description of the colored nodes for Supplementary Figs. S3, S7 and 15 were added to the figure legend.

References

- Jung, M. Y., et al. A hydrophobic ammonia - oxidizing archaeon of the Nitrosocosmicus clade isolated from coal tar - contaminated sediment. *Environ. Microbiol. Rep.* **8**, 983-992 (2016).
- Lücker, S., et al. A Nitrospira metagenome illuminates the physiology and evolution of globally important nitrite-oxidizing bacteria. *Proc. Natl. Acad. Sci.* **107**, 13479-13484 (2010).
- Schröder, C., Elleuche, S., Blank, S., Antranikian, G. Characterization of a heat-active archaeal β -glucosidase from a hydrothermal spring metagenome. *Enzyme Microb. Technol.* **57**, 48-54 (2014).